

**Biogenic silica production and diatom dynamics in the Svalbard region during spring**
Jeffrey W. Krause[1,2], Carlos M. Duarte[3,4], Israel A. Marquez[1,2], Philipp Assmy[5], Mar Fernández-
Méndez[5], Ingrid Wiedmann[6], Paul Wassmann[6], Svein Kristiansen[6], and Susana Agustí[3],
[1]Dauphin Island Sea Lab, Dauphin Island, United States
[2]Department of Marine Sciences, University of South Alabama, Mobile, United States
[3]King Abdullah University of Science and Technology, Thuwal, Kingdom of Saudi Arabia
[4]Arctic Research Centre, Department of Bioscience, Aarhus University, Denmark
[5]Norwegian Polar Institute, Tromsø, Norway
[6]Department of Arctic and Marine Biology, UiT The Arctic University of Norway, Tromsø,
Norway
**Correspondence:** Jeffrey Krause (jkrause@disl.edu)




**Abstract**.
Diatoms are generally the dominant contributors to the Arctic Ocean spring bloom, which is a key
event in regional food webs in terms of capacity for secondary production and organic matter
export.  Dissolved silicic acid is an obligate nutrient for diatoms and has been declining in the
European Arctic.  The lack of regional silicon cycling information precludes understanding the
consequences of such changes for diatom productivity during the Arctic spring bloom.  This study
communicates the results from a cruise in the European Arctic around Svalbard reporting the first
concurrent data on biogenic silica production and export, diatom cellular export, the degree of
kinetic limitation by ambient silicic acid, and diatom contribution to primary production.  Regional
biogenic silica production rates were significantly lower than those achievable in the Southern
Ocean and silicic acid concentration limited the biogenic silica production rate in 95% of samples.
Compared to diatoms in the Atlantic subtropical gyre, regional diatoms are less adapted for silicic
acid uptake at low substrate, and at some stations during the present study, silicon limitation may
have been intense enough to limit diatom growth.  Thus, silicic acid can play a critical role in
diatom spring bloom dynamics.  Diatom contribution to primary production was variable, ranging
from <10% to ~100% depending on the bloom stage and phytoplankton composition.  While there
was agreement with previous studies regarding the rate of diatom cellular export, we observed
significantly elevated biogenic silica export.  Such a discrepancy can be resolved if a higher
fraction of the diatom material exported during our study was modified by zooplankton grazers or
originated from melting ice.  This study provides the most-direct evidence to date suggesting the
important coupling of the silicon and carbon cycles during the spring bloom in the European
Arctic.



## 1    Introduction

Diatoms and the flagellate *Phaeocystis* are the dominant contributors to the Arctic Ocean spring bloom, a cornerstone event supplying much of the annual net community production (Vaquer-Sunyer et al., 2013; Rat'kova and Wassmann, 2002; Wassmann et al., 1999) that fuels Arctic food webs (Degerlund and Eilertsen (2010) and references therein). Hydrographic and chemical changes in the Arctic water column are expected in the future, but whether these will alter diatoms' contribution to spring primary production and organic matter export remains uncertain. Some studies predict lack of ice cover will enhance the spring bloom due to increased light availability (Arrigo et al., 2008), while others predict lower productivity driven by increased stratification and reduced nutrient supply (Tremblay and Gagnon, 2009). Additionally, models predict that warming will lead to a shift from a diatom-dominated bloom to one increasingly dominated by flagellates and picoautotrophs, which has been observed in certain sectors of the Arctic (Li et al., 2009; Lasternas and Agustí, 2010). Because the spring diatom bloom is arguably the single most important productivity event for the Arctic Ocean ecosystem (Degerlund and Eilertsen, 2010; Holding et al., 2015; Vaquer-Sunyer et al., 2013), understanding how diatoms' ecological and biogeochemical importance changes in response to system-wide physical/chemical shifts is important to predict future food web alterations. Diatoms have an obligate requirement for silicon, therefore understanding of regional Si cycling can provide insights into the diatom activity. However, there is a current knowledge gap of regional silicon cycling, which precludes robust assessments of the spring bloom in future scenarios, e.g. Tréguer et al. (2018).

Diatom production is dependent on the availability of dissolved silicic acid ($Si(OH)_4$), which they use to build their shells of biogenic silica ($bSiO_2$). [$Si(OH)_4$] has been observed to be low ($<5\ \mu M$) in the Norwegian Seas and declining over time (Rey, 2012). A more recent analysis demonstrated a decline in pre-bloom [$Si(OH)_4$] concentrations by $1$–$2\ \mu M$ across the north Atlantic subpolar and polar regions over the last 25 years (Hátún et al., 2017); this is consistent with the general Arctic region being a net exporter of silicic acid (Torres-Valdés et al., 2013). This is in stark contrast to the $10$–$60\ \mu M$ [$Si(OH)_4$] observed in the surface waters of the Southern Ocean and the marginal ice zone around Antarctica (Nelson and Gordon, 1982; Brzezinski et al., 2001), where [$Si(OH)_4$] is unlikely to limit the rate of diatom production or biomass yield. Additionally, the stoichiometry of $Si(OH)_4$ availability relative to nitrate (Si:N $<1$) in the source waters, which fuel the spring bloom in most of the north Atlantic and European polar seas, suggests that during a bloom cycle diatoms may experience Si limitation prior to N limitation, especially if diatoms consumed Si and N in near equal quantities as in other diatom bloom regions (Brzezinski et al., 1997; Brzezinski, 1985; Dugdale et al., 1995) and a $2\ \mu M$ threshold [$Si(OH)_4$] defines where diatoms are outcompeted by flagellates (Egge and Aksnes, 1992).

Compared to the Southern Ocean, there is a paucity of field Si-cycling studies in the European Arctic. Reports of diatom silica production are only available from the subarctic northeast Atlantic near $\sim 60°$ N, e.g. between Iceland and Scotland (Allen et al., 2005; Brown et al., 2003), Oslofjorden (Kristiansen et al., 2000), and limited data from Baffin Bay (Hoppe et al., 2018; Tremblay et al., 2002); these previous studies are in zones with higher $Si(OH)_4$ availability than in the European Arctic. Other studies have reported standing stocks of $bSiO_2$ and export in Oslofjorden or the European Arctic, e.g. Svalbard vicinity, Laptev Sea (Hodal et al., 2012; Heiskanen and Keck, 1996; Paasche and Ostergren, 1980; Lalande et al., 2016; Lalande et al., 2013), but none have concurrent measurements of $bSiO_2$ production. Indeed, in the last major review of the global marine silicon cycle, Tréguer and De La Rocha (2013) reported no studies with published $bSiO_2$ production data derived from field measurements from the Arctic.



Currently, we lack a baseline understanding about diatom Si-cycling in the European Arctic
and broader high-latitude north Atlantic region. And while models in the Barents Sea use Si as a
possible limiting nutrient (Wassmann et al., 2006; Slagstad and Støle-Hansen, 1991), there are no
field data to ground truth the modeled parameters governing diatom Si uptake. Thus, there is no
contextual understanding to determine the consequences of the observed changes in regional
[Si(OH)$_4$] since the 1990s and if these affect spring bloom dynamics. This study communicates
the results from a cruise in the European Arctic around Svalbard reporting the first concurrent
datasets on regional bSiO$_2$ production and export, diatom cellular export, and the degree of kinetic
limitation by ambient [Si(OH)$_4$]. Additionally, coupling bSiO$_2$ production rates with
contemporaneous primary production measurements provides an independent assessment for the
diatom contribution to system primary production.
**2      Methods**
**2.1    Region and Sampling**
This study was conducted aboard the RV *Helmer Hanssen* between May 17–29, 2016 as
part of the broader project, ARCEx–The Research Centre for ARCtic Petroleum Exploration
(http://www.arcex.no/). The main goal of this cruise was to study the pelagic and benthic
ecosystem during the Arctic spring bloom around Svalbard and in the northern Barents Sea at
stations influenced by various water masses. The cruise started in the southwestern fjords
influenced by relatively warm Atlantic water, then transited east of Svalbard toward more Arctic-
influenced water (Fig. 1 blue arrow) before turning south towards stations near the Polar Front and
more Atlantic-water station (Fig. 1 red arrows) located to the south of the Polar Front.
Vertical profiles with a CTD were conducted at all stations. Hydrocasts were conducted
using a Seabird Electronics 911 plus CTD with an oxygen sensor, fluorometer, turbidity meter and
PAR sensor (Biospherical/LI-CORR, SN 1060). The CTD was surrounded by a rosette with 12
five-liter Niskin bottles. At two stations, Edgeøya, and Hinlopen, only surface samples were
collected (no vertical profiles with ancillary measurements, Fig. 1). Water was sampled from the
rosette at depths within the upper 40 m (i.e. the extent of the photic layer); for any incubation
described below, the approximate irradiance at the sample depth during collection was mimicked
by placing incubation bottles into a bag made of neutral density screen. Incubation bags were
placed in a deck board acrylic incubator cooled with continuously flowing surface seawater. At
Hinlopen, a block of ice was collected by hand within ~10 m of the vessel and allowed to thaw in
a shaded container for 24 hours at ambient air temperature. After thawing, the melted solution
was homogenized and treated like a water sample for measurement of biomass and rates.
Four sediment trap arrays were deployed between 19 and 23 hours. Arrays in van
Mijenfjorden and Hornsund were anchored to the bottom, whereas the other two arrays (Erik
Erikssenstretet, Polar Front) were quasi-Lagrangian and drifted between 14–16 km during the
deployment. During the Erik Erikssenstretet deployment, the array was anchored to an ice floe.
Arrays included sediment trap cylinders (72 mm internal diameter, 1.8 L volume; KC Denmark)
at 3–7 depths between 20 and 150–200 m, based on bathymetry. After recovery, trap contents
were pooled and subsampled for bSiO$_2$ and phytoplankton taxonomy.
**2.2    Standing stock measurements**
A suite of macronutrients were analyzed at all stations except Hinlopen (just Si(OH)$_4$).
Water was sampled directly from the rosette, filtered (0.7 µm pore size) and immediately frozen.
In the laboratory, nutrients were analyzed using a Flow Solution IV analyzer (O.I. Analytical,



USA) and calibrated with reference seawater (Ocean Scientific International Ltd. UK). Detection limits for $[NO_3 + NO_2]$ and $[Si(OH)_4]$ were 0.02 and 0.07 (µM), respectively. Phosphate was analyzed, but N:P ratios for nutrients were, on average, 8 among all stations, suggesting that N was likely more important than P for primary production. These phosphate data are not discussed.

Samples for biogenic particulates and phytoplankton community composition were taken directly from the rosette and sediment traps. For $bSiO_2$ samples, 600 mL of seawater was collected from the rosette, filtered through a 1.2 µm polycarbonate filter (Millipore); for sediment trap material, less volume was necessary (e.g. 50–100 mL). Most $bSiO_2$ protocols use a 0.6 µm filter cutoff, e.g. Lalande et al. (2016), however, given the magnitude $bSiO_2$ quantified and the size range for regional diatoms we are confident that there was no meaningful systematic underestimate. After filtration, all samples were dried at 60° C and stored until laboratory analysis using an alkaline digestion in Teflon tubes (Krause et al., 2009). For Chl *a*, water-column and sediment samples were collected similarly, filtered on Whatman GF/F (0.7 µm pore size) and immediately frozen (-20 °C). In the laboratory, Chl *a* was extracted in 5 mL methanol in the dark at room temperature for 12 h. The solution was quantified using a Turner Design 10-AU fluorometer, calibrated with Chl *a* standard (Sigma C6144), before and after adding two drops of 5% HCl (Holm-Hansen and Riemann, 1978). Phytoplankton taxonomy and abundance samples were collected in 200 mL brown glass bottles from both the water column and sediment traps, immediately fixed with an aldehyde mixture of hexamethylenetetramine-buffered formaldehyde and glutaraldehyde at 0.1 and 1% final concentration, respectively, as suggested by Tsuji and Yanagita (1981) and stored cool (5°C) and dark. Samples were analyzed with an inverted epifluorescence microscope (Nikon TE300 and Ti-S, Japan), using the Utermöhl (1958) method, in a service laboratory for diatom taxonomy (>90 individual genera/species categories were identified) and abundance at the Institute of Oceanology Polish Academy of Science.

**2.3 Rate measurements**

Biogenic silica production was measured using the radioisotope tracer $^{32}Si$. Approximately 150 or 300 mL samples, depending on the station biomass, were incubated with 260 Bq of high specific activity $^{32}Si(OH)_4$ (>20 kBq µmol Si$^{-1}$). After addition, samples were transported to the deck-board incubator and placed in neutral density screened bags for 24 hours. After incubation, samples were processed immediately by filtering bottle contents through a 25 mm, 1.2 µm polycarbonate filter (Millipore) —matching $bSiO_2$ filtrations. Each filter was then placed on a nylon planchette, covered with Mylar when completely dry, and secured using a nylon ring. Samples were aged into secular equilibrium between $^{32}Si$ and its daughter isotope, $^{32}P$ (~120 days). $^{32}Si$ activity was quantified on a GM Multicounter (Risø National Laboratory, Technical University of Denmark) as described in Krause et al. (2011). A biomass-specific rate (i.e. $V_b$) was determined by normalizing the gross rate ($\rho$) to the corresponding $[bSiO_2]$ at the same depth of collection using a logistic-growth approach (Kristiansen et al., 2000; Krause et al., 2011). For $bSiO_2$ and $\rho$, values within a profile were integrated throughout the euphotic zone (i.e. surface to 1% $I_0$) using a trapezoidal scheme. A depth-weighted $V_b$ was calculated within the euphotic zone by integrating $V_b$ and dividing by the depth integration (Krause et al., 2013).

Two methods were used to assess whether ambient silicic acid $(Si(OH)_4)$ limited diatom Si uptake. The $^{32}Si$ activity additions, incubation conditions, and sample processing are as described above. At four stations (Edgeøya, Polar Front, Hinlopen and Atlantic), eight 300-mL samples collected at a single depth within the euphotic zone and were manipulated to make an eight-point concentration gradient between ambient and +18.0 µM $[Si(OH)_4]$; the maximum concentration



was assumed to saturate Si uptake.  Si uptake has been shown to conform to a rectangular
hyperbola described by the Michaelis-Menten equation:
$$V_b = \frac{V_{max}[Si(OH)_4]}{K_S + [Si(OH)_4]} \quad (1)$$

where $V_{max}$ is the maximum specific uptake rate and $K_S$ is half-saturation constant, i.e.
concentration where $V_b = \frac{1}{2} V_{max}$. Data were fit to the Eq. 1 using a non-linear curve fit algorithm
(SigmaPlot 12.3).  The second type of experiment used only two points: ambient and +18.0 µM
$[Si(OH)_4]$; four-depth profiles were done at three stations (Bellsund Hula, Hornsunddjupet, Erik
Erikssenstretet).  The ratio of Si uptake at +18.0 µM $[Si(OH)_4]$ to Si uptake at ambient $[Si(OH)_4]$
defines an enhancement (i.e. Enh) statistic.  This two-point approach was conducted at all depths
in the euphotic zone; Enh ratios >1.08 imply kinetic limitation beyond analytical error given the
methodology (Krause et al., 2012).
Net primary productivity (PP) was quantified concurrently with biogenic silica production
at six stations at the depth of approximately 50% of surface irradiance (Table 1). Carbon uptake
rates were measured using a modification of the $^{14}$C uptake method (Steemann Nielsen, 1952).
Water samples were spiked with 0.2 µCi mL$^{-1}$ of $^{14}$C labelled sodium bicarbonate (Perkin Elmer,
USA) and distributed in three clear and one dark plastic bottles (40 mL each). Subsequently, they
were incubated for 24 h in the deck incubator with a 50% light reduction mesh. After incubation,
samples were filtered onto 0.2 µm nitrocellulose filters. The filters were stored frozen (-20 °C) in
scintillation vials with 10 ml EcoLume scintillation liquid (MP Biomedicals LLC, USA) until
further processing. Once on land, the particulate $^{14}$C was determined using a scintillation counter
(TriCarb 2900 TR, Perkin Elmer, USA). The carbon uptake values in the dark were subtracted
from the mean of the triplicate carbon uptake values measured in the light incubations. Using
contemporaneous ρ measurements and PP measurements, the diatom contribution to PP is
estimated as:
$$\text{Diatom \%PP} = 100 \times \frac{\rho \times (Si:C)^{-1}}{PP} \quad (2)$$

where the Si:C ratio for diatoms can be used from culture values, e.g. 0.13 (Brzezinski, 1985).
Export rates were calculated using the standing stock measurements, length of deployment,
and trap opening area.  These approaches are common and detailed elsewhere (Wiedmann et al.,
2014; Krause et al., 2009).
**3     Results**
**3.1   Hydrography and Spatial patterns**
The regional ecosystem around Svalbard is driven by ice dynamics (Sakshaug, 2004). One
week prior to the cruise, a majority of the southern Svalbard archipelago had open water, which
was anomalous compared to similar dates in previous years (e.g. 2014, 2015, ice data archived at
http://polarview.met.no/).  By the end of the cruise, Svalbard could have been entirely circled by
the vessel, with only open drift ice in the northeastern region.  While 2016 was among the lowest
years for total Arctic sea ice, the ice extent in Svalbard and the Barents Sea is highly dynamic.  Ice
edges may be pushed southward into the Barents Sea proper by wind while areas to the north
remain ice free, e.g. Wassmann et al. (1999) and references therein.
Spatial patterns in hydrography and nutrients were highly variable.  In the southwestern
stations (e.g. fjords and Atlantic-influenced water), the surface temperature ranged between 1–4
°C; similar temperature was observed in the Atlantic station south of the Polar Front (Fig. 1E).
Northeastern domain stations were more influenced by Arctic water and the surface temperatures
ranged between -2–1 °C (Fig. 1E).  Surface nutrient concentrations, particularly $[NO_3+NO_2]$ and



[Si(OH)$_4$], showed a broad range. The highest surface [NO$_3$+NO$_2$] was observed in the
southwestern fjords, 2–>8 µM, and the Atlantic station (~3 µM, Fig. 1A). The surface
concentrations at the remaining stations were <0.5 µM or near detection limits (Fig. 1A).
[Si(OH)$_4$] was lower than [NO$_3$+NO$_2$] (i.e. Si:N <1) among stations where [NO$_3$+NO$_2$] was > 0.1
µM. At high [NO$_3$+NO$_2$] stations, the [Si(OH)$_4$] ranged from 1.1–4.5 µM (Fig. 1B) but the range
was lower among other stations (0.4–1.1 µM, Fig. 1B). bSiO$_2$ (proxy for diatom biomass, Fig.
1C) was typically similar to, or lower than, surface [Si(OH)$_4$]. The highest surface [bSiO$_2$] was
observed in the southern stations (Atlantic-influenced waters), ~ 2–3 µmol Si L$^{-1}$ (Fig. 1C). At
most other stations the [bSiO$_2$] was <1 µmol Si L$^{-1}$. Among all stations/depths bSiO$_2$ varied by a
factor of ~40 (does not include Hinlopen ice algae).
Primary productivity, measured at six stations at 5 m (approximately 50% of surface
irradiance), varied over two orders of magnitude. The lowest rates were observed at the four
stations having lowest surface [NO$_3$+NO$_2$] and ranged from 2–13 µg C L$^{-1}$ d$^{-1}$; at these stations
[Chl $a$] ranged from 2.0–4.8 µg L$^{-1}$ (Table 1, Fig. 1D). The highest rates were measured at van
Mijenfjorden and Bredjupet, 100 ±65 µg C L$^{-1}$ d$^{-1}$ and 27 ±1 µg C L$^{-1}$ d$^{-1}$, respectively, and
corresponded to high [NO$_3$+NO$_2$] and low [Chl $a$] 1.8 and 0.7 µg L$^{-1}$, respectively (Table 1, Fig.
1D).

## 3.2 Vertical profiles

As expected, most stations showed strong vertical gradients in nutrient concentrations.
Profiles in the southwestern region of Svalbard (van Mijenfjorden, Bredjupet) had elevated
[Si(OH)$_4$], with little vertical structure. Vertical [Si(OH)$_4$] profiles among other stations showed
typical nutrient drawdown between the surface and ~20 m. At these stations, surface [Si(OH)$_4$]
concentrations were typically <1.5 µM and subsurface values (to 20 m) ranged from 0.5–3.0 µM
(Fig. 2A). [NO$_3$+NO$_2$] exceeded [Si(OH)$_4$] among all depths at five stations (van Mijenfjorden,
Bredjupet, Hornsund, Atlantic; Fig. 2B), whereas in the remaining stations [NO$_3$+NO$_2$] exceeded
[Si(OH)$_4$] (i.e. Si:N <1) at depths >5 m (Bellsund Hula), >20 m (Erik Erikssenstretet) and >27 m
(Polar Front). For the latter three stations, [NO$_3$+NO$_2$] had a significant drawdown in surface
waters, but then increased with depth without a similar degree of vertical enhancement in
[Si(OH)$_4$] (Fig. 2).
[bSiO$_2$] was typically highest at or near the surface, with a maximum of ~2 µmol Si L$^{-1}$
(Fig. 2C). At the Bellsund Hula and Erik Erikssenstretet stations, subsurface [bSiO$_2$] maxima were
present (Fig. 2C; note–no surface data are available for van Mijenfjorden). Among non-profile
stations, [bSiO$_2$] was within the range observed among vertical profiles except for the Hinlopen
ice algae, where the melt water had exceptionally high [bSiO$_2$] (Fig. 2C). The surface-to-20-m
integrated bSiO$_2$ (∫bSiO$_2$) spanned over an order of magnitude, with a low at Bredjupet (1.9 mmol
Si m$^{-2}$) and a high at Hornsunddjupet (42.4 mmol Si m$^{-2}$, Table 1) despite their proximity (~50
km).
Diatom abundance and taxonomy data were sampled at fewer stations, but the vertical and
spatial variability generally mirrored trends in [bSiO$_2$]. In the surface waters of van Mijenfjorden
and Hornsund, diatom abundances ranged between 5x10$^4$–5x10$^5$ cells L$^{-1}$ in the upper 50 m (Fig.
3A). However, within the same vertical layer at the Erik Erikssenstretet and Polar Front (duplicate
profiles) stations, diatom abundances were enhanced by up to two orders of magnitude (4x10$^4$–
4x10$^7$ cells L$^{-1}$, Fig. 3A). When integrated to 40-m depth (∫Diatom), matching the shallowest
sediment-trap depth among the three stations reported (Fig. 3E–H), diatom inventories also

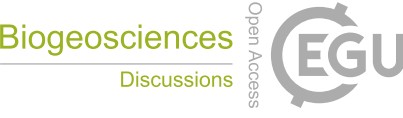



showed a two-order of magnitude variability as observed in ∫bSiO₂. ∫Diatom was lowest at van
Mijenfjorden ($7.67 \times 10^9$ cells m$^{-2}$) and highest at Polar Front station ($527 \times 10^9$ cells m$^{-2}$, Table 1).
Among the stations which had corresponding sediment trap deployments (van
Mijenfjorden, Hornsund, Erik Erikssenstretet), the diatom-assemblage composition was similar
despite differences in abundance. The van Mijenfjorden station was dominated by *Thalassiosira*
(e.g. *T. antarctica*, *T. gravida*, *T. hyalina*, *T. nordenskioeldii*), *Fragilariopsis cylindrus*, and
*Chaetoceros furcellatus* (Fig. 3B). *Chaetoceros* spp. was nearly absent from Erik Erikssenstretet
(Fig. 3D) and of little importance at Hornsund (Fig. 3C). *Thalassiosira* species (same as van
Mijenfjorden) cells also dominated Hornsund and Erik Erikssenstretet among most depths (Fig.
3C, D). However, at Hornsund, deeper depths were dominated by diatom groups less frequently
observed ("Other diatom" category, Fig. 3), and with small contributions from *Fragilariopsis*
*cylindrus* and *Navicula vanhoefenii*.
Diatom bSiO₂ productivity, ρ, mirrored trends in biomass. Among the profiles, rates
generally varied from ρ <0.01 to 0.11 µmol Si L$^{-1}$ d$^{-1}$ (Fig. 2D). ρ was highest in the Atlantic
station (Fig. 2D), which was expected given the higher bSiO₂ (Fig. 2C). However, the rates in the
Hinlopen ice algae were like those quantified at Hornsunddjupet, ~ 0.1 µmol Si L$^{-1}$ d$^{-1}$, despite the
ice algae station having an order of magnitude more biomass. This suggests the Hinlopen ice algae
were senescent or stressed and a sizable portion of the measured bSiO₂ was non-active or detrital.
When integrated in the upper 20-m, ∫ρ ranged from 0.27 - 1.46 mmol Si m$^{-2}$ d$^{-1}$ (Table 1), which
is a smaller proportional range than observed in ∫Diatoms and ∫bSiO₂. Overall, bSiO₂-normalized
rates ($V_b$) were low among all stations and depths (<0.01 to 0.13 d$^{-1}$). The depth-weighted $V_b$, i.e.
$V_{AVE}$, had a narrower range between 0.03–0.13 d$^{-1}$. Thus, doubling times for bSiO₂ in the upper
20 m ranged between 5–23 days.
The rate of diatom biogenic silica production was reduced by ambient [Si(OH)₄] in 95% of
the samples examined. Full kinetic experiments verified that Si uptake conformed to Michaelis-
Menten kinetics (Fig. 4A; adjusted R$^2$ ranged 0.64–0.92 among experiments). The highest $V_{max}$
was observed in the Atlantic station (0.36 ±0.02 d$^{-1}$), which also had the highest ambient [Si(OH)₄]
among the full kinetic experiments (1.4 µM). $V_{max}$ observed at Edgeøya and the Polar Front were
nearly identical (0.05 ±<0.01 d$^{-1}$ for both) and lowest in the Hinlopen ice diatoms (0.02 ±<0.01 d$^{-1}$
). $K_S$ constants had a narrower range, with a low of 0.8 ±0.3 µM at the Polar Front and between
2.1–2.5 µM among the other three stations. Among these full-kinetic experiments, the Enh ratio
ranged from 1.8–7.7 with the most intense [Si(OH)₄] limitation of uptake observed in the Hinlopen
ice diatoms. For profiles where two-point kinetic experiments were conducted, the same trends
were observed (Fig. 4B). The Enh ratio was similar among depths at Bellsund Hula (1.5–2.2),
Hornsunddjupet and Bredjupet (3.4–5.4 for latter two stations, Fig. 4B). At Erik Erikssenstretet,
Enh ratios were more variable, ranging from 2.8–7.3 in the upper 10 m with no Enh effect (i.e.
<1.08) observed at 20 m —this was the only sample and depth which showed no resolvable degree
of kinetic limitation for Si uptake.
Rates of bSiO₂- and diatom export were variable. Among the three sediment trap regions,
bSiO₂ export rates ranged from ~4–10 mmol Si m$^{-2}$ d$^{-1}$ (Fig. 2E). These rates are significant and
represent up to 50% of the ∫bSiO₂ in upper 20 m at van Mijenfjorden (Table 1). For diatom cells,
a similar degree of variability was observed. Export at van Mijenfjorden ranged from 390–1500
$\times 10^6$ cells m$^{-2}$ d$^{-1}$, similar ranges to Hornsund (520–2800 $\times 10^6$ cells m$^{-2}$ d$^{-1}$) and Erik Erikssenstretet
(510–860 $\times 10^6$ cells m$^{-2}$ d$^{-1}$, Fig. 3E). The Atlantic station had significantly higher diatom export
(800–2300 $\times 10^6$ cells m$^{-2}$ d$^{-1}$) among all depths in the upper 120 m (Fig. 3E). The bSiO₂ and
diatom cellular export were highly correlated (r = 0.67, p<0.01; Spearman's Rho Test). Among




all stations, *Fragilariopsis cylindrus* had the highest contribution to diatom export, and
*Thalassiosira* species (e.g. *T. antarctica*, *T. gravida*, *T. hyalina*, *T. nordenskioeldii*) were also
important (Fig. 3F-H). In Hornsund, *Navicula (N. vanhoefenii, N. sp.)* was an important group for
export (Fig. 3G) but this was not observed elsewhere. Similarly, "Other diatom" groups were
proportionally important at Erik Erikssenstretet (Fig. 3H), as were *Thalassiosira* resting spores at
the Atlantic station (data not shown). Among all diatoms, the only groups which were numerically
important in both the water column and the sediment traps were *Fragilariopsis cylindrus* and
*Thalassiosira* species (Fig. 3B–D, F–H).

## 4   Discussion
### 4.1   Diatom Si cycling relative to other systems

To our knowledge, this is the first report of $bSiO_2$ production data of the natural diatom
community in this sector of the Arctic. Other studies have reported $\rho$ data in the subarctic Atlantic
Ocean (Brown et al., 2003; Kristiansen et al., 2000; Allen et al., 2005) ~10–20° latitude south of
our study region or in Baffin Bay (Hoppe et al., 2018; Tremblay et al., 2002). However, the Hoppe
et al. (2018) study only includes $\rho$ measured after a 24-hour manipulation experiment and only at
one site and depth near the Clyde River just east of Nunavat (Canada), no data are reported for the
ambient conditions, and the measurements from Tremblay et al. (2002) are based on net changes
in standing stocks instead of gross $bSiO_2$ production. Banahan and Goering (1986) report the only
$\rho$ to date in the southeastern Bering Sea; however, Varela et al. (2013) recently reported that
$[Si(OH)_4]$ in surface waters (>5 µM) are unlikely to be significantly limiting to diatoms in any
sector of the Bering, Chukchi or Beaufort Sea regions. Around Svalbard, some previous studies
have examined other Si-cycling components including variability in $bSiO_2$ in the water column
(Hodal et al., 2012) and sediments (Hulth et al., 1996), $bSiO_2$ and diatom export (Lalande et al.,
2016; Lalande et al., 2013), or trends in $[Si(OH)_4]$ (Anderson and Dryssen, 1981). The $\rho$
measurements presented here have no straight forward study for comparison; therefore, we
compare these to the previous high-latitude Atlantic data and to well-studied sectors of the
Southern Ocean.
During our study, $\int\rho$ in the Svalbard vicinity was low. Working in the NE Atlantic between
Iceland and Scotland, Brown et al. (2003) reported $\int\rho$ between 6–166 mmol Si $m^{-2}$ $d^{-1}$. In the same
region, under post-bloom conditions, Allen et al. (2005) reported 7 mmol Si $m^{-2}$ $d^{-1}$ for one profile.
These rates are significantly higher than at out four profile stations (Table 1), and the degree of
difference does not appear to be driven by differences in integration depth (compared to our study,
Table 1). Given the higher $[Si(OH)_4]$ in the southern region of the Atlantic subpolar gyre (Hátún
et al., 2017), the maximum achievable $\int\rho$ may vary with latitude. While our profile sampling was
opportunistic, it appears we sampled some stations with significant diatom biomass (high $\int bSiO_2$),
but the corresponding production rates ($\int\rho$) were low, with estimated doubling times on the order
of 11–23 days. This suggests these high-biomass stations may have been near, or past, peak bloom
conditions (Fig. 2A, B) and the seasonal timing is consistent with regional field and modeling
studies inferring diatom bloom dynamics from Chl *a* trends, e.g. (Wassmann et al., 2010; Oziel et
al., 2017). Kristiansen et al. (2000) reported $\rho$ in Oslofjorden during the late winter (February–
March), rates ranged from 0.03–2.0 µmol Si $L^{-1}$ over nine sampling periods with corresponding
$V_b$ between <0.01–0.28 $d^{-1}$; however, this system has a higher $Si(OH)_4$ supply and surface
concentrations at the start of the bloom period were >6 µM, approximately 50% higher than the
highest surface concentrations observed during our study (Fig. 2A). Nearly all the initial $Si(OH)_4$
was eventually converted to $bSiO_2$ during the bloom (Kristiansen et al., 2001; Kristiansen et al.,



2000). The specific rates observed in our study fall within the lower values reported by Kristiansen
et al. (2000), which may be explained by the reduced uptake from lower [Si(OH)$_4$] (e.g. Fig. 4).
The Southern Ocean is one of the most globally significant regions for production of bSiO$_2$.
The surface [Si(OH)$_4$] and [NO$_3$+NO$_2$] are among the highest in the ocean and the source waters
usually have >50% excess Si(OH)$_4$ relative to nitrate (Brzezinski et al., 2002). Thus, exceptional
Si(OH)$_4$ drawdown relative to nitrate is required for diatom biomass yield to be limited by Si in
this region. The mean ∫ρ in sectors of the Southern Ocean are variable. In the Weddell Sea, winter
rates range between 2.0–3.2 mmol Si m$^{-2}$ d$^{-1}$ in the seasonal ice zone (Leynaert et al., 1993).
Within the sub-Antarctic zone, rates averaged 1.1 and 4.8 mmol Si m$^{-2}$ d$^{-1}$ in the summer and
spring, respectively (Fripiat et al., 2011). At the terminus of diatom blooms in the sub-Antarctic
and polar frontal zone, rates can be lower, e.g. 0.1–0.3 mmol Si m$^{-2}$ d$^{-1}$ (Fripiat et al., 2011); such
values are similar to the range observed during our study, especially since these Southern Ocean
studies integrated ∫ρ deeper than 40 m (e.g. 50–100 m). Brzezinski et al. (2001) reported average
∫ρ ~25 mmol Si m$^{-2}$ d$^{-1}$ (integrated from surface to 80–120 m) during intense blooms in the seasonal
ice zone which propagated south of the Antarctic polar front. But despite the massive diatom bSiO$_2$
accumulating in these blooms, V$_{AVE}$ generally ranged between 0.05–0.15 d$^{-1}$ (Brzezinski et al.,
2001). Given the order-of-magnitude difference in [Si(OH)$_4$] and ∫ρ between the Arctic and
Southern Ocean, the similar V$_{AVE}$ in both regions may be more reflective of thermal effects on
diatom growth rate, since Si uptake and diatom growth rates are tightly coupled, or a significant
accumulation of detrital bSiO$_2$ (i.e. diatom fragments) in the Southern Ocean, where low
temperatures reduce bSiO$_2$ remineralization rates (Bidle et al., 2002).
**4.2    Potential for Silicon limitation of diatom productivity**
Suboptimal silicon availability affects the rate of diatom bSiO$_2$ production and can limit
their growth. A widely cited [Si(OH)$_4$] threshold, below which diatoms will be outcompeted by
other phytoplankton, is ~2.0 µM; this metric was derived from a comparison of diatom abundance
(relative to total microplankton) versus [Si(OH)$_4$] during mesocosm experiments in a Norwegian
fjord system (Egge and Aksnes, 1992). Applying this metric globally has been criticized due to
observation of diatom dominance among microplankton when [Si(OH)$_4$] <1 µM in systems
ranging from fjords to the open-ocean (Krause et al., 2013; Hodal et al., 2012; Kristiansen et al.,
2001) and also culture studies showing some diatom species can maintain high growth rates when
[Si(OH)$_4$] <0.5 µM (reviewed by Kristiansen and Hoell (2002)). Stoichiometry of silicon
availability relative to nitrate also help diagnose Si limitation; the most widely accepted diatom
Si:N ratio is ~1 based on temperate and low-latitude clones (Brzezinski, 1985). There is a paucity
of diatom culture studies examining stoichiometry in polar diatoms, but Si:N during spring blooms
in Oslofjorden are close to Brzezinski's Si:N ratio (Kristiansen et al., 2001). For diatoms in
Svalbard and the broader region of the subpolar and polar European Atlantic, both [Si(OH)$_4$] and
its availability relative to N appear to be suboptimal for creating intense diatom blooms, such as
those occurring in the Southern Ocean. Yet, the Arctic spring bloom is consistently dominated by
diatoms or *Phaeocystis* (Degerlund and Eilertsen, 2010), which suggests some level of adaptation
for diatoms to the low [Si(OH)$_4$] environment of the region.
Nutrient relationships support the potential for silicon to be a controlling factor of regional
diatom productivity. When plotting [NO$_3$+NO$_2$] as a function of [Si(OH)$_4$] (Fig. 5A) a few trends
emerge: 1) The slope of the linear regression relationship (2.5 ± 0.1 mol N (mol Si)$^{-1}$) denotes that
NO$_3$+NO$_2$ is consumed at over twice the rate per unit Si(OH)$_4$. 2) Given that the source water
[NO$_3$+NO$_2$] concentration is only ~twice that of [Si(OH)$_4$], a 2.5 drawdown ratio would predict





$NO_3+NO_2$ to be depleted before $Si(OH)_4$. This indeed indicates that phytoplankton can deplete
nitrogen to levels below detection while they appear unable to deplete $Si(OH)_4$ pools below 0.5
µM, which would indicate 0.5 µM is the ultimate $Si(OH)_4$ concentration required to support diatom
growth. Nitrate and silicic acid drawdown within the upper 50 m during the spring season (1980–
1984) was discussed by Rey et al. (1987) who suggested apparent nitrate limitation (1980, 1981)
and silicic acid limitation (1983, 1984) are annually variable. The Reigstad et al. (2002) analysis
of nitrate and silicic acid drawdown in the central Barents Sea shows similarities in that the diatom
assemblage could only drawdown $[Si(OH)_4]$ to ~1 µM (May 1998) and ~0.5 µM (July 1999).
These authors suggest that physical effects on phytoplankton explain the interannual variability in
the maximum $[Si(OH)_4]$ drawdown, where diatoms dominate in shallow mixed waters opposed to
*Phaeocystis pouchetii* dominating in deeper mixed waters.
None of these nitrate and silicic acid relationships capture the progressive dynamics of an
active diatom bloom. Using the ARCEx data (Fig. 5A), if diatoms are limited by an absolute
$[Si(OH)_4]$ (e.g. 2 µM), then at this concentration there is still ample residual $[NO_3+NO_2]$ (3.8 µM)
which could be used by other phytoplankton that do not consume Si (Fig. 5A). Even if the diatom
$[Si(OH)_4]$ threshold is closer to 1 µM, this observation of excess $[NO_3+NO_2]$ (1.4 µM) still holds.
Diatoms have an *r*-selected ecological strategy and are typically the first phytoplankton group to
bloom in this region under stratified shallow-mixed conditions (Reigstad et al., 2002). If they
consumed N and Si in near equal amounts (i.e. Si:N ~1) without significant competition for N by
other major phytoplankton groups, it is highly probable that Si would limit them first during a
bloom. Clearly, interannual and local differences in mixing, which may favor *Phaeocystis
pouchetii* over diatoms (Reigstad et al., 2002), can affect the assemblage and nutrient drawdown
trajectory (e.g. see points with high $[Si(OH)_4]$ and little measurable $[NO_3+NO_2]$, Fig. 5A);
therefore, diagnosing whether Si could limit diatom growth requires additional analyses.
When considering the European sector of the Arctic/sub-Arctic between 60°–80° N, there
is compelling evidence that $[Si(OH)_4]$ limits the rate of diatom $bSiO_2$ production. During ARCEx,
the relationship between $V_b$ and $[Si(OH)_4]$ also supports that Si regulates diatom productivity to
some degree. Our kinetic data demonstrate that in three of four experiments $K_S$ was ~2.0 µM, but
in the Polar Front the $K_S$ was lower ~0.8 µM. These data are consistent with community kinetic
experiments reported in Oslofjorden where $K_S$ and $V_{max}$ were between 1.7–11.5 µM and 0.16–0.64
$d^{-1}$, respectively, with the lowest $V_{max}$ observed during the declining diatom bloom (Kristiansen et
al., 2000). These authors concluded that silicon ultimately controlled diatom productivity during
this bloom (Kristiansen et al., 2001). In the only other kinetic experiment reported in the northeast
Atlantic, Allen et al. (2005) observed a linear response in $V_b$ between ambient and 5 µM $[Si(OH)_4]$,
which suggests uptake did not show any degree of saturation at this concentration. These field-
based $K_S$ values are considerably higher than parameters used in Barents Sea models, e.g. 0.5 µM
(Slagstad and Støle-Hansen, 1991), 0.05 µM (Wassmann et al., 2006) which reflect the high
efficiency uptake seen in culture (Paasche, 1975). Fitting a regression to the $V_b V_{max}^{-1}$ as a function
of $[Si(OH)_4]$ (line shown in Fig. 5B) suggests that 2.3 µM is the best constrained half-saturation
concentration (i.e. concentration where $V_b V_{max}^{-1} = 0.5$) for the regional assemblage; however, this
is biased from the Hornsunddjupet assemblage (white symbols, Fig. 5B), and this aggregated half-
saturation would increase to 2.8 µM if those data were not considered. Unlike diatoms in the north
Atlantic Subtropical Gyre, e.g. Sargasso Sea (Krause et al., 2012), regional diatoms do not appear
well-adapted for maintaining $V_b V_{max}^{-1} > 0.5$ at low $[Si(OH)_4]$. Instead, diatoms during the spring
season appear to be best adapted for concentrations exceeding 2.3 µM, which suggests that as
$[Si(OH)_4]$ is depleted diatoms may slow growth (Fig. 5B).



To avoid growth limitation, diatoms can reduce their silicon per cell when [Si(OH)$_4$] is suboptimal. An accepted principal from culture work is that diatoms can alter their silicon per cell by a factor of four (Martin-Jézéquel et al., 2000). Thus, when uptake is reduced to <25% of $V_{max}$, diatoms must slow growth to take up enough Si to produce a new cell. Using the empirical half-saturation constant range (2.3–2.8 µM) calculated from Fig. 5B and using Eq. 1 to solve for the concentrations where $V_b\ V_{max}^{-1} \leq 0.25$ ($V_{max}$ is a constant), suggests that when [Si(OH)$_4$] is below 0.3–0.8 µM, the degree of kinetic limitation could force diatoms to slow growth in response. Such a range could be biased low given the influence of the highly efficient Hornsunddjupet assemblage (which was associated with warmer Atlantic waters). But at these [Si(OH)$_4$] there would also be up to 0.8 µM [NO$_3$ + NO$_2$] remaining (Fig. 5A). Therefore, under shallow stratified conditions which favor diatoms over *Phaeocystis* (*sensu* Reigstad et al. (2002)), [Si(OH)$_4$] may regulate regional diatom productivity during spring consistent with similar results from southern-Norway fjords (Kristiansen et al., 2001). This provides the most direct assessment to date supporting the general idea that Si may limit regional diatom productivity (Rey, 2012; Rey et al., 1987; Reigstad et al., 2002).

### 4.3 Diatom contribution to primary production

Among the six sites with paired PP and ρ measurements, the bloom phase can be inferred from the magnitude of nutrient drawdown, [Chl *a*], PP, and pCO$_2$ (data not shown). Bredjupet appeared to be a pre-bloom station given the nutrients, while the van Mijenfjorden station also appeared to be in an early bloom phase. The Erik Erikssenstretet station represented a peak bloom condition, whereas assemblages at Hornsunddjupet and Edgeøya appeared to be post bloom and in a stage of decline. The Polar Front station represented the end or late-phase bloom condition; however, at this station *Phaeocystis* was abundant (data not shown), suggesting it may have dominated the bloom dynamics instead of diatoms.

The diatom contribution to PP was highly variable. Among the stations with high [NO$_3$ + NO$_2$] (van Mijenfjorden, Bredjupet) the diatom contribution to PP (e.g. Eq. 2) was low, 5–6%. At two stations, Hornsunddjupet and the Polar Front, the diatom contribution to PP increased to 48–57%. In the Edgeøya, and Erik Erikssenstretet stations, diatoms accounted for all PP, 130 and 340%, respectively. Such unrealistic value at Erik Erikssenstretet could imply a potential issue with the Si:C ratio (Eq. 2), specifically an increase in Si per cell and/or lower C per cell due to reduced growth rate associated with the peak/end of the bloom.

Clearly, diatoms can play a significant role in local productivity, but these data demonstrate a "bloom and bust" nature. At stations at or near peak bloom levels (e.g. Edgeøya, Erik Erikssenstretet), diatoms could account for nearly all primary production. However, they may also conduct an insignificant percentage of primary production prior to the onset of the bloom (e.g. van Mijenfjorden, Bredjupet). But even when physical conditions may favor *Phaeocystis* blooms, diatoms appear to be significant contributors to primary production (Polar Front station). In such a situation, N would be predicted to be the limiting nutrient as it will be consumed by both *Phaeocystis* and diatoms whereas Si will only be consumed by the latter.

In the European Arctic, shifts in summer-period phytoplankton communities away from diatom-dominated conditions have been observed in numerous studies. One of the dominant features has been the increasing abundances of Phaeocystis in ice-edge (Lasternas and Agustí, 2010) or under-ice blooms (Assmy et al., 2017). These changes have corresponded with larger-scale shifts in the export of diatoms to depth in the Fram Straight (Nöthig et al., 2015; Lalande et al., 2013; Bauerfeind et al., 2009). The timing of these shifts, e.g. mid-2000s, correspond with the

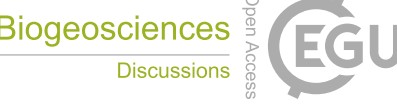



broader regional reduction in winter mixed-layer [Si(OH)$_4$] concurrent with the shift to negative gyre-index state in the latter half of the decade (Hátún et al., 2017). With a reduction in pre-bloom Si(OH)$_4$ supply, diatoms may run into limitation sooner and thus leave more residual nitrate for non-diatom phytoplankton. Degerlund and Eilertsen (2010) also demonstrate a dynamic temperature niche for individual diatom groups/species. Coello-Camba et al. (2015) showed that temperature induced a shift in the Arctic phytoplankton community, with diatoms declining as temperature increased and thereby favoring dominance of flagellates. Given the highly variable contribution of diatoms to primary productivity in this system in spring and the effects which carry over into summer, should climate change or natural physical oscillations affect diatoms in this system, resolving such a signal will be challenging. A similar conclusion about detecting a climate-change signal was made in the eastern Bering Sea by Lomas et al. (2012) given the natural variability in primary production.

### 4.4 Diatoms and export

The bSiO$_2$ export rates observed during ARCEx were significant relative to the standing stocks. At van Mijenfjorden, the rate of export in the upper 40 m represented 39% of the ∫bSiO$_2$ standing stock (23.3 mmol Si m$^{-2}$) in the same vertical layer. This quantity was much higher than at Erik Erikssenstretet, where the 40-m export rate was <11% of the ∫bSiO$_2$ in the upper water column (note: no samples were taken deeper than 20 m, thus, additional bSiO$_2$ between 20–40 m would lower the 11% estimate). Given that the van Mijenfjorden site was located within shallow fjord waters (bottom depth approximately 60 m), such a high proportion export relative to standing stock may suggest either lateral focusing processes (e.g. discussed by DeMaster (2002)) and/or resuspension of sediment bSiO$_2$ into the water and resettlement. The rate of bSiO$_2$ export was also at least a factor of four higher than ∫ρ in the upper 20 m. It is likely that some fraction of ∫ρ was missed due to lack of sampling between 20–40 m, but with a lack of light at these depths, it is unlikely systematic underestimates of ρ caused the disparity. Given the deeper water at the Erik Erikssenstretet and Atlantic stations, such high bSiO$_2$ export may be driven by previously high ρ and bSiO$_2$ standing stock which accumulated in the overlying waters or, given the dynamic circulation in the region, this signal may have been laterally advected to these station locations.

Relative to previous studies, the bSiO$_2$ export rates were also high. During May 2012 in Kongsfjorden, Lalande et al. (2016) reported bSiO$_2$ export rates between 0.2–1.3 mmol Si m$^{-2}$ d$^{-1}$ in the upper 100 m, a similar range was observed by Lalande et al. (2013) in the eastern Fram Strait using moored sediment traps (2002–2008) collecting at depths between 180–280 m. Lalande et al. (2013) concluded that, despite warm anomaly conditions, pulses of bSiO$_2$ export were positively correlated to the presence of ice in the overlying waters which stratifies the water and helps initiate a diatom bloom. However, if the light was insufficient to stimulate a bloom, Lalande et al. (2013) suggested much of the pulse of bSiO$_2$ exported to depth may have originated in the ice and sank during melting. Indeed, the low V$_b$ (<0.01 d$^{-1}$) observed at the Hinlopen station (ice algae), despite the moderate ρ measured (0.12 µmol Si L$^{-1}$ d$^{-1}$), suggests that most of the ice-associated bSiO$_2$ was detrital and not associated with living diatoms. Thus, the recent ice retreat observed prior to the ARCEx cruise was a potential source of such high bSiO$_2$ export to depth despite the considerably lower ∫ρ in the upper 20 m.

Among the groups examined, the most important diatom genera for standing stock and export were *Thalassiosira* and *Fragilariopsis*, suggesting these groups are important drivers of bulk bSiO$_2$ fluxes. Given the large-size and chain-forming life histories for the dominant species within each genus, it is likely that their dominance in the trap abundances helps explain the high





correlation (r = 0.67, p<0.01; Spearman's Rho Test) between $bSiO_2$ and diatom export. Given this
degree of correlation, it would be expected that both $bSiO_2$ and diatom export would be similarly
enhanced relative to previous studies; however, this was not observed. Diatom cellular export in
Kongsfjorden (Lalande et al., 2016) were similar-to or a factor of three lower than rates quantified
during ARCEx (Table 1, Fig. 3E), whereas $bSiO_2$ export during ARCEx was over an order of
magnitude higher than $bSiO_2$ export in Kongsfjorden. One possible explanation for the higher
degree of $bSiO_2$ export enhancement, relative to diatom cellular export, between studies is that
more exported material during ARCEx was modified in the food web. For instance, in Erik
Erikssenstretet gel traps confirm the presence of aggregates and mesozooplankton fecal pellets
(Wiedmann et al. in prep), and in van Mijenfjorden detrital particles were most prominent on the
gel traps opposed to clearly recognizable material (e.g. diatom valves). These observations suggest
the potential for considerable modification of diatom organic matter prior to export (diatoms in
fecal pellets, fragments associated with aggregates, etc.). This is consistent with previous
observation in the Barents Sea showing high potential for copepod fecal pellets to be exported in
the Polar Front and Arctic-influenced regions during spring (Wexels Riser et al., 2002). And
supports the general ideas for the importance of diatom organic matter in fueling secondary
production regionally during this season (Degerlund and Eilertsen (2010) and references therein).
**4.5   Conclusion**
This is the first regional data set with contemporaneous measurements of diatom $bSiO_2$
standing stock, production, export and assessment of kinetic limitation by $[Si(OH)_4]$ in the
European Arctic. Among stations and depths there was widespread limitation of diatom $bSiO_2$
production rates by ambient $[Si(OH)_4]$ during spring-bloom conditions. The kinetic parameters
for diatom Si uptake (e.g. $K_S$) quantified in our study are significantly higher than rates used in
regional models and quantified in polar diatom cultures; therefore, these data will help future
modeling efforts better simulate diatom/Si dynamics. Given the trajectories of Si and N
consumption, diatom-dominated blooms (vs. *Phaeocystis*-dominated) could deplete $Si(OH)_4$ prior
to nitrate; and at some stations, the degree of kinetic limitation by ambient $[Si(OH)_4]$ could have
resulted in diatom growth being slowed. Diatom contribution to PP was highly variable, ranging
from <10% to ~100% depending on the bloom stage; but even when *Phaeocystis* appeared to be
favored, diatoms still had a significant (~50%) contribution to PP. While there was agreement
with previous regional studies regarding the rate of diatom cellular export, we observed
significantly elevated $bSiO_2$ export. Such a discrepancy can be resolved if a higher fraction of the
diatom material exported during our study was modified by zooplankton grazers, relative to
previous studies, or if much of this $bSiO_2$ was derived from melting ice and/or advection.
*Data availability.* All data are available upon request to the authors or are available through the
UiT research data bank (https://dataverse.no/dataverse/uit).
*Author contributions.* JK, CMD, SA conceived/designed the study and conducted analysis. JK,
CMD, IM, PA, MFM, IW, SA conducted the fieldwork. PW and SK conducted analysis. All co-
authors contributed to the writing of the paper, led by JK.
*Competing interests.* The authors declare that they have no conflict of interest.




*Acknowledgments.* The authors thank the science party and crew of the RV *Helmer Hanssen*. We also thank S. Øygarden, E. Kube, A. Renner, D. Vogedes, H. Foshaug, S. Acton, D. Wiik, B. Vaaja and W. Dobbins for logistic support. Primary data analysis was supported by the Dauphin Island Sea Lab. Vessel time, ancillary data and I. Wiedmann's and P. Wassmann's contribution was supported by ARCEx, funded by industry partners and the Research Council of Norway (project #228107). P. Assmy was supported by the Research Council of Norway (project no. 244646) and M. Fernández-Méndez by the Ministry of Foreign Affairs, Norway (project ID Arctic). J. Krause, C. Duarte, and S. Agustí were supported by internal funding sources at their respective institutions.

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





Table 1 – Station properties including surface temperature, nutrients and chlorophyll a (± standard deviation), 20-m biogenic silica stock (∫bSiO$_2$),
production (∫ρ) and depth-weighted specific production (V$_{AVE}$), 40-m integrated diatom abundance (∫Diatom) and export of bSiO$_2$ and diatoms at
40 m. The disparity between the integration depths for bSiO$_2$ standing stock and diatom abundance reflects the lack of bSiO$_2$ samples to 40 m
depth and that the latter are used to compare with diatom export (Discussion). Note: Hinlopen (ice) station not included. The Polar Front ∫Diatom
is the mean of two profiles.

| Station Name | T (°C) | [NO$_3$ + NO$_2$] (µM) | [Si(OH)$_4$] (µM) | [Chl $a$] (µg L$^{-1}$) | 20-m ∫bSiO$_2$ (mmol Si m$^{-2}$) | 20-m ∫ρ (mmol Si m$^{-2}$ d$^{-1}$) | 20-m V$_{AVE}$ (d$^{-1}$) | 40-m ∫Diatom abundance (10$^9$ cells m$^{-2}$) | 40-m bSiO$_2$ export (mmol Si m$^{-2}$ d$^{-1}$) | 40-m Diatom export (10$^6$ cells m$^{-2}$ d$^{-1}$) |
|---|---|---|---|---|---|---|---|---|---|---|
| ‡van Mijenfjorden | -0.43 | 8.1 | 3.8 | 1.84 ±0.19 | 10.8 | - | - | 7.67 | 9.03 | 769 |
| ‡Bredjupet | 4.72 | 9.4 | 4.5 | 0.72 ±0.03 | 1.9 | 0.27 | 0.13 | - | - | - |
| Bellsund Hula | 0.69 | <0.1 | 0.5 | 2.66 ±0.05 | 15.3 | 0.49 | 0.06 | - | - | - |
| Hornsund | -0.28 | 1.6 | 1.1 | 2.50 ±0.20 | - | - | - | 8.97 | - | 1180 |
| ‡Hornsunddjupet | -0.20 | <0.1 | 0.4 | 2.43 ±0.17 | 42.2 | 1.46 | 0.03 | - | - | - |
| ‡Edgeøya | -0.70 | 0 | 0.7 | 1.99 ±0.03 | - | - | - | - | - | - |
| ‡Erik Erikssenstretet | -1.58 | 0.4 | 0.4 | 4.77 ±0.31 | 34.9 | 1.03 | 0.04 | 252 | 4.00 | 436 |
| ‡*Polar Front Station | 2.19 | <0.1 | 1.1 | 3.00 ±0.03 | - | - | - | 527 | - | - |
| Atlantic | 4.10 | 3.3 | 1.4 | 6.66 ±0.33 | - | - | - | - | 9.20 | 2380 |

†Surface value
*25 m depth
‡Denotes concurrent primary production and biogenic silica production measurements at one depth






Figure Captions:

Figure 1: Surface properties during 2016 ARCEx cruise including A) Nitrate + Nitrite (µM), B) dissolved
silicic acid (µM), C) biogenic silica (µmol Si $L^{-1}$), D) Chlorophyll $a$ (µg $L^{-1}$) and E) Temperature (°C)
overlaid on station map.  Station names are denoted on the map and colored arrows generalize the flow of
Atlantic-influenced (red) and Arctic-influenced (blue) waters.

Figure 2:  Vertical profiles for A) dissolved silicic acid, B) Nitrate + Nitrite, C) biogenic silica standing
stock, D) biogenic silica production rate, and E) biogenic silica export.  Symbols are associated by station,
and line connectors are used to denote profile data opposed to individual symbols noting samples at one
depth.
Figure 3:  Diatom abundance (A) and assemblage composition (B–D) in the water column, and diatom
export (E) and assemblage composition (F–H) within sediment traps.  Note – taxonomy information only
shown for stations where both water-column and sediment-trap data were available (see text for species).
Resting spores (e.g. *Chaetoceros*, *Thalassiosira*) were absent from the 40-m sediment traps; thus,
proportional abundances for spore-producing taxa are entirely for vegetative cells.  For panel A, there are
replicate diatom abundance measurements (from separate hydrocasts) for the Polar Front station.

Figure 4:  Assessment of Si uptake limitation by available silicic acid during ARCEx.  A) 8-point kinetic
experiments taken at four stations (legend next to panel B).  Data were fit to a Michaelis-Menten
hyperbola using SigmaPlot 12.3 software.  B) Enh. ratio profiles (i.e. $V_b$ in +18.0 µM $[Si(OH)_4]$ treatment
relative to $V_b$ in the ambient $[Si(OH)_4]$ treatment) at four stations.
Figure 5: Diagnosis of potential silicon limitation for diatom production during ARCEx.  A) Nitrate +
Nitrite drawdown as a function of dissolved silicic acid.  B) The ratio of $V_b$ at ambient $[Si(OH)_4]$ to $V_{max}$
versus dissolved silicic acid.  In both panels, linear regressions were done using a Model II reduced major
axis method.  For comparison, the same relationship for the Sargasso Sea in the North Atlantic
subtropical gyre, as synthesized in Krause et al. (2012).



Figure 1:

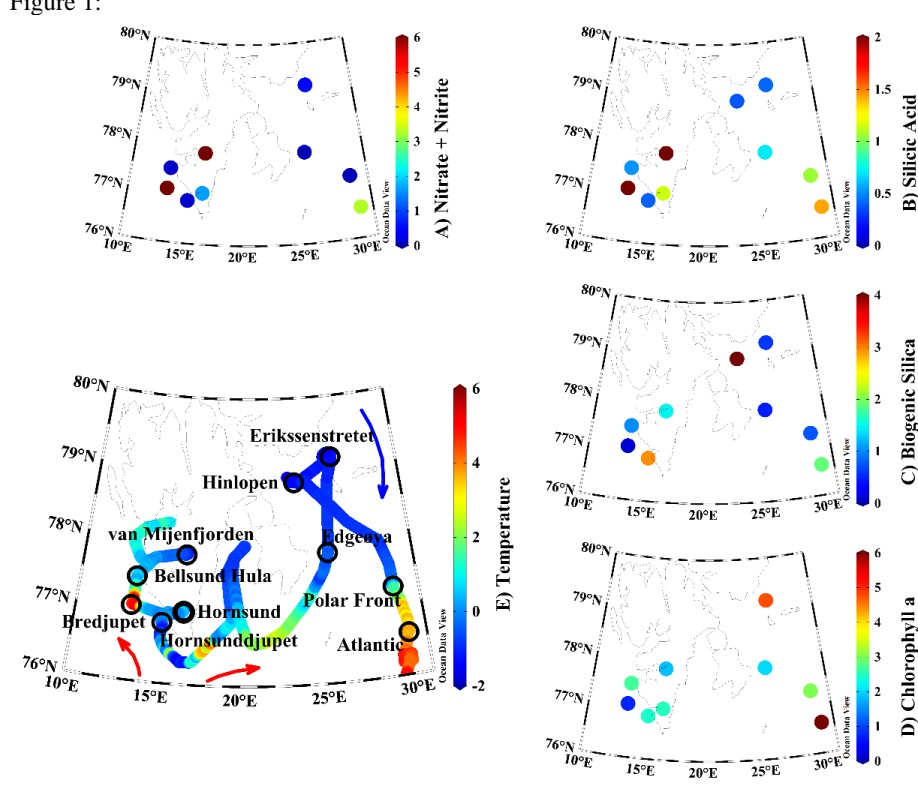




Figure 2:

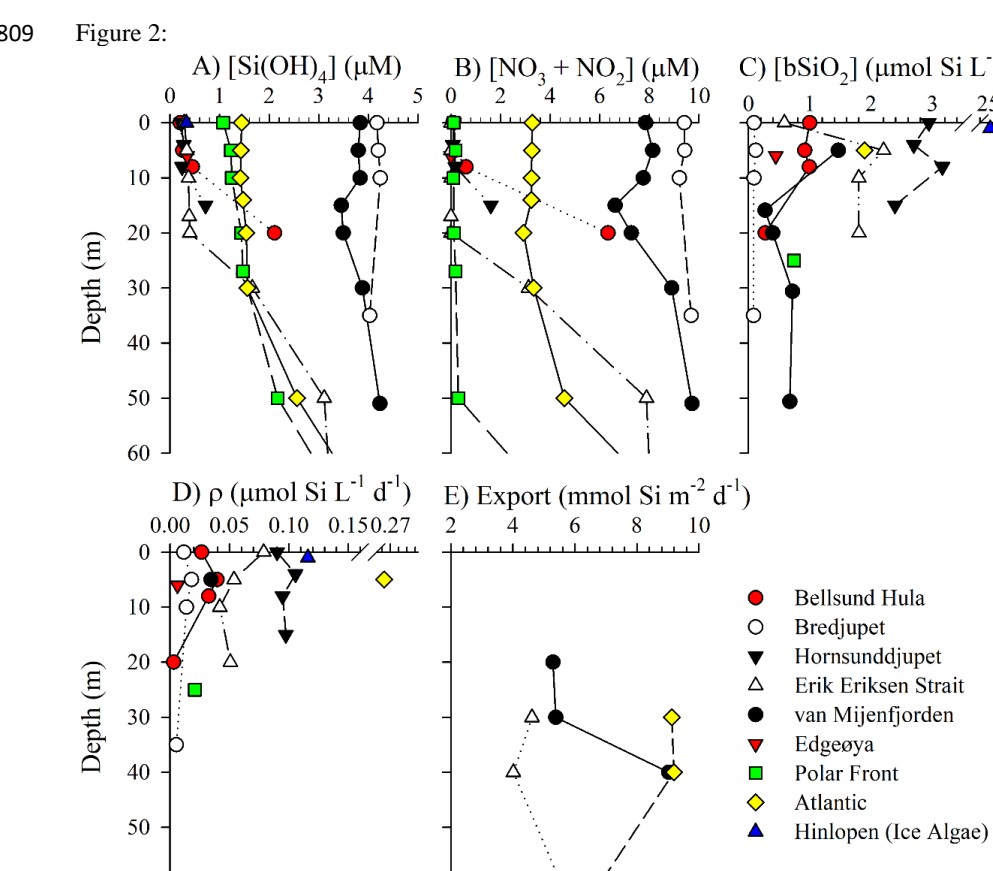




Figure 3:

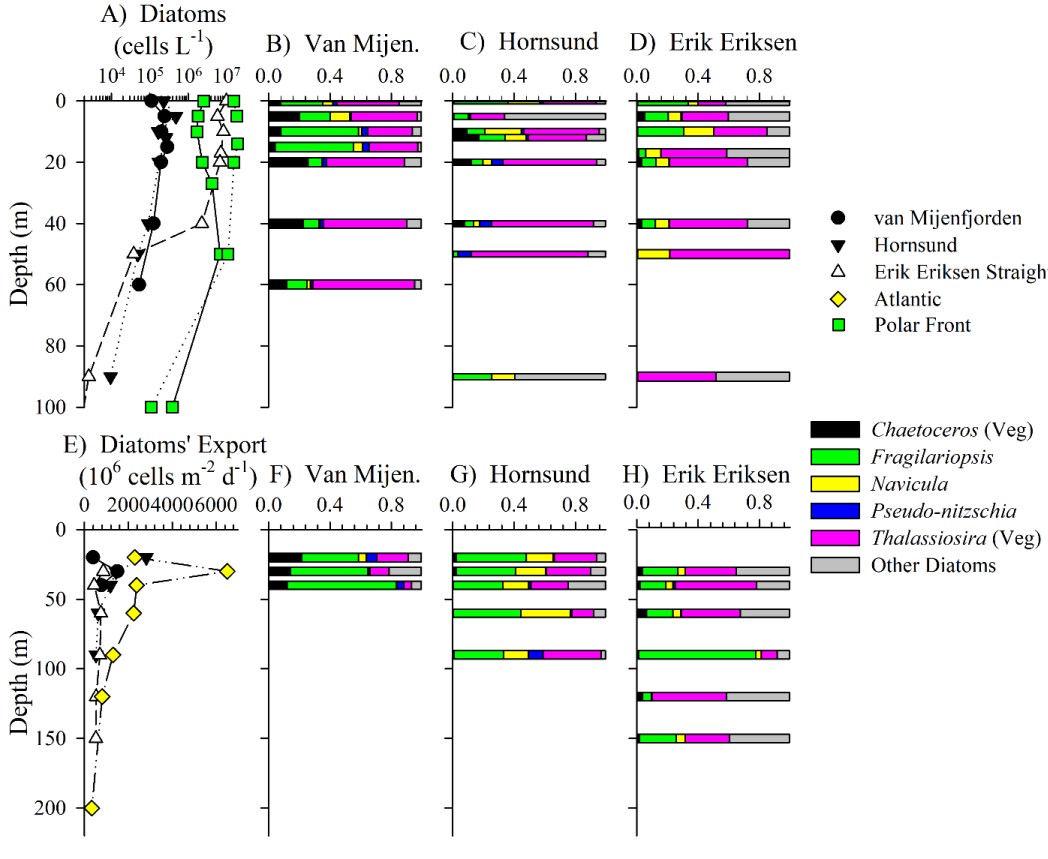




Figure 4:

A)

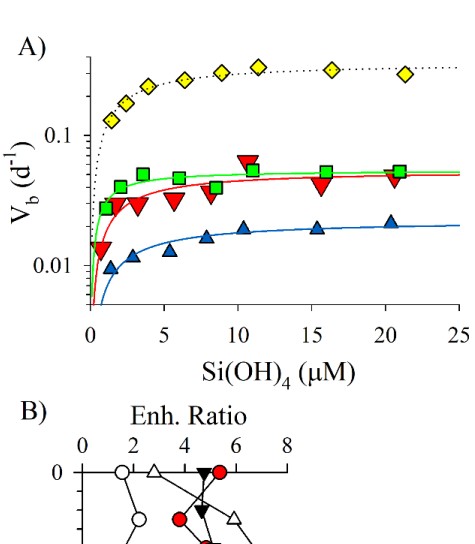

B)

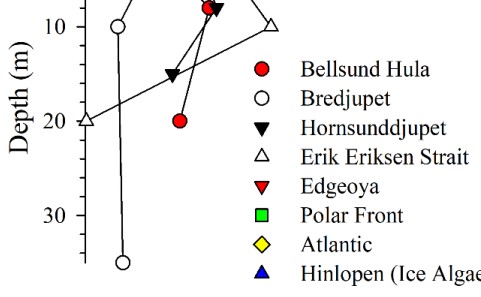






Figure 5:

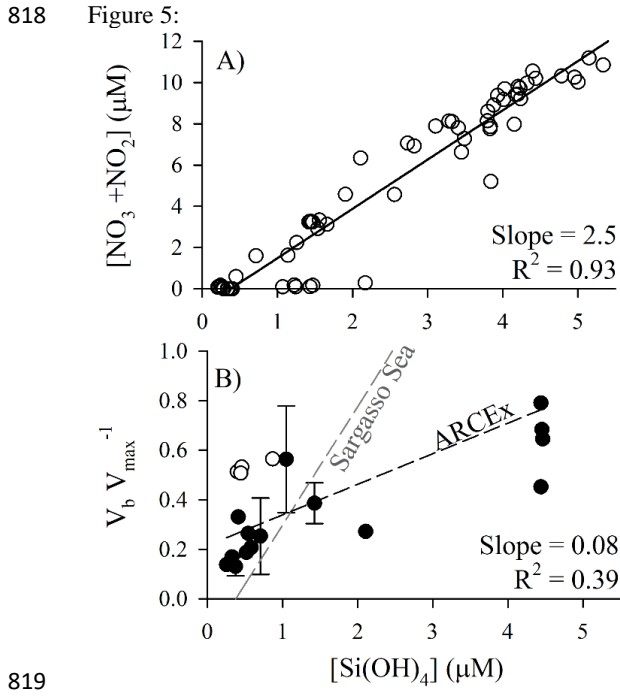
