# Peer review of "Biogenic silica production and diatom dynamics in the Svalbard region during spring Jeffrey W. Krause1,2, Carlos M. Duarte3,4, Israel A. Marquez1,2, Philipp Assmy5, Mar Fernández-Méndez5, Ingrid Wiedmann6, Paul Wassmann6, S"

_Biogeosciences, 2018_

## Referee Comment (RC1) · Anonymous Referee #1 · 11 Jun 2018

The manuscript of Krause *et al.* presents interesting results concerning biogenic silica production and export levels as well as estimates of kinetic constants from an opportunistic sampling near the Svalbard Archipelago in the Arctic Ocean by late spring. The data presented are the first direct (silicon–32 method) measurements of biogenic silica production in the Arctic Ocean, which in themselves deserve publication. From these data, the authors then attempt to establish the potential control of diatom production by the availability of silicic acid as well as the contribution of diatoms to total primary production. However, I think the authors are pushing their limited data set much too far and that the manuscript should be shortened by getting more concise.

• Firstly, I find it difficult to understand why the authors focus on the so-called Egge & Aksnes 2 µM $H_4SiO_4$ threshold value, as it is clear that the data from this publication have been wrongly interpreted in several past publications (which is recognized by the authors besides). I suggest just using their kinetic values to discuss the potential limitation of diatom uptake by $H_4SiO_4$ availability, and then shortly discuss hypotheses for growth limitation, which is another point not directly assessed in this study. ON the other hand, authors might consider that the actual limitation starts under 2 times $K_S$. The comparison between nitrate and silicic acid concentrations is not very clear. First, they mention a 2.5 slope (Figure 5) between nitrate and silicic acid, which means that nitrate is taken up 2.5 times faster than silicic acid, and then authors take a 1:1 ratio to discuss the potential for silicic acid limitation. I can imagine that they are trying to decipher the relative contribution of siliceous vs. non-siliceous (e.g. *Phaeocystis*) components of the phytoplankton community but this should be clearly indicated. On top of that, the use of a 1:1 Si:N ratio is questionable (large species–specific variations, see below for Si:C).

• The contribution of diatoms to primary production is also another weak point of the ms. The calculation is based on a transformation of ρSi to ρC by using the average Brzezinski' ratio of 0.13. This is a very simplistic way of addressing this important question as this ratio is known to be subject to large species–specific variations (e.g. Brzezinski gives a Si:C biomass range of 0.15 ± 0.04 for large diatoms, which could result in a ~2 times range for ρC estimates). This is somehow risky business and should, at least, be acknowledged and discussed.

• Finally, the authors present data for direct diatom cell export but the underlying issue is not clearly stated: Do they want to compare direct diatom sedimentation by mass sinking to other export vectors such as repackaging? If yes this should be clearly stressed.

line 28 : "diatom cellular export" – the wording is misleading (could be export from a diatom cell). I'd rather use "export of diatom cells".

line 65 : " A more recent analysis demonstrated a decline in pre-bloom [Si(OH)₄] concentrations by 1–2 µM across the north Atlantic subpolar and polar regions over the last 25 years (Hátún *et al.*, 2017); this is consistent with the general Arctic region being a net exporter of silicic acid (Torres-Valdés *et al.*, 2013)." – I don't see the consistence between the decrease of $H_4SiO_4$ concentrations and the net exportation of this nutrient; please rephrase.

line 68 : " This is in stark contrast to the 10–60 µM [Si(OH)₄] observed in the surface waters of the Southern Ocean and the marginal ice zone around Antarctica (Nelson and Gordon, 1982; Brzezinski et al., 2001), where [Si(OH)₄] is unlikely to limit the rate of diatom production or biomass yield." – I disagree; There are ample references to state that actually the reverse is true, due to sometimes unusual high $K_S$ (e.g. Nelson & Tréguer MEPS 1992, Nelson et al. DSR II 2001, Mosseri et al. DSR II 2008).

line 76 : " ... and a 2 µM threshold [Si(OH)₄] defines where diatoms are outcompeted by flagellates (Egge and Aksnes, 1992)." – I strongly disagree with that sentence. The work

of Egge and Aksnes did not evidence any real threshold (no kinetic values measured) and just merely indicated areas of realized niches for diatom vs. flagellates with regards to Si vs. P availability. Please do not cite this reference in such a way that was even not addressed by the authors of this paper. + as indicated above.

line 137 : "… suggesting that N was likely more important than P for primary production." – As authors refer to absolute concentrations, the correct phrasing should be: "… suggesting that N was likely more important than P for potentially limiting primary production."

line 138 : "These phosphate data are not discussed." – Even though a range would be welcome.

line 153 : "… fixed with an aldehyde mixture of hexamethylenetetramine-buffered formaldehyde and glutaraldehyde at 0.1 and 1% final concentration, respectively, as suggested by Tsuji and Yanagita (1981) …" – although this should be OK this is not the usual fixative for diatoms (acidic Lugol preferred), partly due to its toxicity for the microscopical examiner.

line 164 : " … neutral density screened bags …" – please mention the photometric levels used.

line 207 : "Export rates were calculated using the standing stock measurements, length of deployment, and trap opening area." – Please give the model/type of sediment trap.

line 260 : "… except for the Hinlopen ice algae, where the melt water …" – Is that naturally–melted ice or meltwater produced by ice melting in the lab? Please clarify.

Line 331 : " Brown $et$ $al.$, 2003 " – Comment: For some strange reason L. Brown's incubations lasted for only 6 hours, which renders her production results questionable.

line 339 : "… Varela $et$ $al.$ (2013) recently reported that $[Si(OH)_4]$ in surface waters (>5 µM) are unlikely to be significantly limiting to diatoms in any sector of the Bering, Chukchi or Beaufort Sea regions." – Although for Subarctic waters Brown $et$ $al.$ (2003, mentioned just above) kinetic experiments show a strong limitation (non-saturating kinetics) up to 30 µM.

line 387 : " Suboptimal silicon availability affects the rate of diatom $bSiO_2$ production and can limit their growth. A widely cited $[Si(OH)_4]$ threshold, below which diatoms will be outcompeted by other phytoplankton, is ~2.0 µM; this metric was derived from a comparison of diatom abundance (relative to total microplankton) versus $[Si(OH)_4]$ during mesocosm experiments in a Norwegian fjord system (Egge and Aksnes, 1992)." – should be removed: No need to discuss this threshold as it is mentioned that it is strongly criticized (and see my comment above).

line 410 : " This indeed indicates that phytoplankton can deplete nitrogen to levels below detection while they appear unable to deplete $Si(OH)_4$ pools below 0.5 µM, which would indicate 0.5 µM is the ultimate $Si(OH)_4$ concentration required to support diatom growth." – I disagree with this interpretation. The 0.5 µM Si level just reflects the residual $H_4SiO_4$ stock after complete removal of nitrate.

line 422 : "… if diatoms are limited by an absolute $[Si(OH)_4]$ (e.g. 2 µM), …" – This is speculative: By what evidence is this proposition supported?

line 436 : "… the relationship between $V_b$ and $[Si(OH)_4]$ also supports that Si regulates diatom productivity to some degree." – The large dispersion of data points on Figure 5 results in a very weak relationship, so that there is certainly something else explaining the low realized $V_b$ at the 4.5 µM $H_4SiO_4$ level.

line 443 : " ... Allen *et al.* (2005) observed a linear response in Vb between ambient and 5 μM [Si(OH)$_4$], which suggests uptake did not show any degree of saturation at this concentration." – Also in Brown *et al.* (2003); as mentioned above.

line 517 : " At van Mijenfjorden, the rate of export in the upper 40 m represented 39% of the ∫bSiO2 standing stock (23.3 mmol Si m-2) in the same vertical layer." – I don't understand as from Table 1 it seems that the standing stock is 10.8 and the export 9.03?

line 524 : " The rate of bSiO$_2$ export was also at least a factor of four higher than ∫ρ in the upper 20 m." – I was not able to find where did this come from.

---

## Referee Comment (RC2) · Anonymous Referee #2 · 19 Jul 2018

**Review on Biogenic silica production and diatom dynamics in the Svalbard region during spring by Krause et al.**

Krause et al. investigated phytoplankton, especially diatoms, and nutrients at 9 stations in the Atlantic sector north of 76°N. They measured silicate, nitrate plus nitrite, chlorophyll a, biogenic silica, determined diatom assemblage, estimate productivity and export (based on sediment traps). The silicic acid concentration in the upper 50 m was always below 5 $\mu$mol L$^{-1}$ and at most stations below the nitrate plus nitrite concentration. At several stations [Si(OH)$_4$] was below 1 or even 0.5 $\mu$mol L$^{-1}$ in the upper 20 m. In order to investigate Si uptake limitation, the authors performed on board growth experiments over a range of [Si(OH)$_4$] at 4 stations. Michaelis-Menten functions for silicic acid uptake (Eq.1) were fit to the data yielding estimates for maximum uptake rates ($V_{\mathrm{max}}$) and half-saturation constants ($K_{\mathrm{S}}$). Let me suggest listing these estimates in a table. The estimates for $K_{\mathrm{S}}$ are much higher than some estimates (for different diatom species) reported in the literature (for example, Paasche, 1973a,b), however, lower than the higher values given by Kristiansen et al. (2000). What might explain this large range and these differences? Could it be influenced by factors (other nutrients, grazing) differing between the various investigations/experiments? The manuscript contains valuable new data, is well written, and will be of interest to many readers of Biogeosciences. I recommend publication after minor revisions.

**Further remarks/suggestions**:
L123-129 Description of trap deployment was not very detailed ('at 3-7 depths between 20 and 150-200 m, based on bathymetry'). It would be good to add a list with depths and bottom depths (or a reference where to find this information).
L132-134 Freezing sample for nutrient analysis: Procedures (thawing, measurements how long after thawing) and quality control (freeze certified reference material in parallel with samples) have been discussed in the literature (for example, Macdonald et al., 1986, Clementson & Wayte, 1992, Dore et al., 1996). Could you please give more details on the procedures and quality control?
L136-138 "Phosphate was analyzed, but N:P ratios for nutrients were, on

average, 8 among all stations, suggesting that N was likely more important than P for primary production." N:P is below Redfield thus N might be limiting primary production before P. However, 'N was likely more important than P for primary production' sounds strange. Please rewrite.

L145,148,225 60° C → 60°C, -20 °C → -20°C, -2-1 °C → -2 to +1°C (no gaps; please check whole manuscript)

L175 please rewrite "dividing by the depth integration" → dividing by depth-integrated values

L199 ml → mL

L205-206 using C:Si (instead of Si:C) would avoid the exponent -1 in Eq.(2) and give values more in Redfield-style, i.e. molar Si:C = 0.13 → 7.7 C:Si (only slightly higher than the Redfield C:N). What's the uncertainty of the Si:C estimate?

L295-296 "The rate of diatom biogenic silica production was reduced by ambient $[Si(OH)_4]$ in 95% of the samples examined." sounds strange. I guess you mean 'was kinetically limited by ambient $[Si(OH)_4]$' based on comparison with estimated $K_S$ values or based on enhancement factors.

L317,548 Spearman's Rho Test: add number of data n = ...

L380-384 What about grazing?

**References**

[1] Clementson, Lesley A. and Wayte, Sally E. The effect of frozen storage of open-ocean seawater samples on the concentration of dissolved phosphate and nitrate. *Water Research*, 26(9):1171–1176, 1992.

[2] Dore, J.E., T. Houlihan, D.V. Hebel, G. Tien, L. Tupas, and D.M. Karl. Freezing as a method of sample preservation for the analysis of dissolved inorganic nutrients in seawater. *Marine Chemistry*, 53(3-4):173–185, 1996.

[3] Macdonald, RW and McLaughlin, FA and Wong, CS. The storage of reactive silicate samples by freezing. *Limnology and Oceanography*, 31(5):1139–1142, 1986.

[4] Paasche, E. Silicon and the ecology of marine plankton diatoms. I. *Thalassiosira pseudonana* (*Cyclotella nana*) grown in a chemostat with silicate as limiting nutrient. *Marine biology*, 19(2):117–126, 1973a.

[5] Paasche, E. Silicon and the ecology of marine plankton diatoms. II. Silicate-uptake kinetics in five diatom species. *Marine Biology*, 19(3):262–269, 1973b.

---

## Author Comment (AC1) · 20 Aug 2018

Below, we address comments from RC1. We cluster each comment and separate them as "1" Reviewer comments, "2,3" are responses and revisions. A pdf version (bg-2018-226-RC1-supplement_Krause_etal_response) of this response has been uploaded in the supplement.

1) The manuscript of Krause et al. presents interesting results concerning biogenic silica production and export levels as well as estimates of kinetic constants from an opportunistic sampling near the Svalbard Archipelago in the Arctic Ocean by late spring. The data presented are the first direct (silicon–32 method) measurements of biogenic silica production in the Arctic Ocean, which in themselves deserve publication. From

these data, the authors then attempt to establish the potential control of diatom production by the availability of silicic acid as well as the contribution of diatoms to total primary production. However, I think the authors are pushing their limited data set much too far and that the manuscript should be shortened by getting more concise. 2,3) We thank the reviewer for their constructive criticism, these have strengthened the manuscript. Below, we address the concerns. The revisions are aimed to both address these issues and make the manuscript more concise.

1) Firstly, I find it difficult to understand why the authors focus on the so-called Egge & Aksnes 2 $\mu$M H4SiO4 threshold value, as it is clear that the data from this publication have been wrongly interpreted in several past publications (which is recognized by the authors besides). 2,3) We agree with the reviewer and the focus on this apparent threshold value is due to its wide use in the field. As of August 2018, the Egge & Aksnes (1992) manuscript has been cited over 700 times, and many explicitly regard this 2 $\mu$M threshold for diatom microplankton dominance among many systems. This type of prose has been reduced; however, given this is the threshold embraced by the community it must be acknowledged.

1) I suggest just using their kinetic values to discuss the potential limitation of diatom uptake by H4SiO4 availability, and then shortly discuss hypotheses for growth limitation, which is another point not directly assessed in this study. ON the other hand, authors might consider that the actual limitation starts under 2 times KS. 2,3) The original version discusses kinetic limitation between lines 434 to 455, followed by a section discussing the possibility that silicic acid may be low enough to limit diatom growth (line 459 – 462). If we interpret the reviewer correctly, then "under 2 times Ks" (i.e. half of the half-saturation constant) is the silicic acid concentration when uptake (i.e. Vb) is 25% of maximum uptake (i.e. Vmax) which is the same metric we use to diagnose potential growth limitation (line 461). We have made this clearer (e.g. denote the equivalence of "under 2 times Ks" and Vb/Vmax <0.25) in the revision.

1) The comparison between nitrate and silicic acid concentrations is not very clear.

First, they mention a 2.5 slope (Figure 5) between nitrate and silicic acid, which means that nitrate is taken up 2.5 times faster than silicic acid, and then authors take a 1:1 ratio to discuss the potential for silicic acid limitation. I can imagine that they are trying to decipher the relative contribution of siliceous vs. non-siliceous (e.g. Phaeocystis) components of the phytoplankton community but this should be clearly indicated. On top of that, the use of a 1:1 Si:N ratio is questionable (large species–specific variations, see below for Si:C). 2,3) The approach silicic acid versus nitrate drawdown has been used elsewhere (Monterey Bay, Brzezinski et al. 1997; Barents Sea, Rey et al. 1987; Southern Ocean diatom cultures, Takeda 1998). We agree with the reviewer that this nutrient data includes drawdown of nitrate from other organisms, likely Phaeocystis, hence the slope exceeds 1 (i.e. nitrate drawn down faster than silicic acid). We also agree that there is variability within the drawdown ratios. However, there is currently a lack of relevant culture data to apply for the conditions observed here (we know of recent manuscripts, currently in preparation, to address this gap; but those cannot be cited formally here). Most polar culture work focuses on Fe-limitation (e.g. Takeda 1988, Nature) and Southern Ocean clones; however, under Fe-replete conditions Takeda (1998) reported Si:N between 0.73 (Chaetoceros dichaeta) and 1.2 (Nitzschia sp.), the average among these clones is 0.97, not unlike the 1:1 ratio from Brzezinski 1985. In the revision, we make this point with more clarity and provide the Takeda (1998) study as a literature basis for the 1:1 ratio (beyond the canonical Brzezinski stoichiometry).

1) The contribution of diatoms to primary production is also another weak point of the ms. The calculation is based on a transformation of rSi to rC by using the average Brzezinski' ratio of 0.13. This is a very simplistic way of addressing this important question as this ratio is known to be subject to large species–specific variations (e.g. Brzezinski gives a Si:C biomass range of 0.15 $\pm$0.04 for large diatoms, which could result in a $\sim$2 times range for rC estimates). This is somehow risky business and should, at least, be acknowledged and discussed. 2,3) As with the previous point, we agree that there is variability within the stoichiometry. While admittedly simplistic, we also have coarse resolution (n = 6 coupled measurements), which (in our opinion) do

not merit as sophisticated approach given it would not be robust; this data coupling was serendipitous (hence the low n value) but do provide a first-order estimate. The estimates cluster into three modes when using Si:C of 0.13, i.e. diatoms insignificant (line 482), diatoms ∼50% of primary production (line 483-484), diatoms ∼all primary production (lines 484-485). We point out the simplicity of this ratio for the Erik Erikssen-stretet station (line 486-487) which could be due to stoichiometric changes associated with growth rate shifts (likely modifying Si:C) during a bloom phase. To this section, we added a minor addendum stating that if we are off by a factor of two in Si:C (i.e. up to 0.26), then the resulting clusters would change (i.e. 2 stations where diatoms were insignificant, 2 stations where diatoms did ∼25% of production, 1 station where diatom did ∼50% of production, 1 station where diatoms did ∼100% of production). However, this does not change the two main interpretations: 1) diatoms quantitative contribution to primary production is "bloom and bust" and 2) even when diatom biomass is relatively low, their contribution to production can still quantitatively important (e.g. 25 – 50%).

1) Finally, the authors present data for direct diatom cell export but the underlying issue is not clearly stated: Do they want to compare direct diatom sedimentation by mass sinking to other export vectors such as repackaging? If yes this should be clearly stressed. 2,3) The reviewer's interpretation of our intent is correct, we suggest that the reason why cellular export is similar to previous studies, despite biogenic silica export being higher than other studies, is consistent with repackaging which we termed "food web effect." We have revised this for clarification.

1) line 28 : "diatom cellular export" – the wording is misleading (could be export from a diatom cell). I'd rather use "export of diatom cells". 2,3) Changed as suggested.

1) line 65 : " A more recent analysis demonstrated a decline in pre-bloom [Si(OH)4] concentrations by 1–2 $\mu$M across the north Atlantic subpolar and polar regions over the last 25 years (Hátún et al., 2017); this is consistent with the general Arctic region being a net exporter of silicic acid (Torres-Valdés et al., 2013)." – I don't see the consistence between the decrease of H4SiO4 concentrations and the net exportation of this nutrient; please rephrase. 2,3) This is rephrased, e.g. removing "this is consistent with the general Arctic region being a net exporter of silicic acid (Torres-Valdés et al., 2013)"

1) line 68: " This is in stark contrast to the 10–60 $\mu$M [Si(OH)4] observed in the surface waters of the Southern Ocean and the marginal ice zone around Antarctica (Nelson and Gordon, 1982; Brzezinski et al., 2001), where [Si(OH)4] is unlikely to limit the rate of diatom production or biomass yield." – I disagree; There are ample references to state that actually the reverse is true, due to sometimes unusual high KS (e.g. Nelson & Tréguer MEPS 1992, Nelson et al. DSR II 2001, Mosseri et al. DSR II 2008). 2,3) The reviewer is correct as kinetic limitation (i.e. ambient silicic acid limits the rate of diatom silica production) is clearly observed in the Southern Ocean. Our intent (which was not clearly conveyed) was to contrast the effects of the high silicic acid in the Southern Ocean vs. the Arctic. This is rephrased "... where [Si(OH)4] is unlikely to limit diatom growth unless iron is replete, and stimulates exceptional blooms which consume Si, or assemblages are highly inefficient for Si uptake."

1) line 76: " ... and a 2 $\mu$M threshold [Si(OH)4] defines where diatoms are outcompeted by flagellates (Egge and Aksnes, 1992)." – I strongly disagree with that sentence. The work of Egge and Aksnes did not evidence any real threshold (no kinetic values measured) and just merely indicated areas of realized niches for diatom vs. flagellates with regards to Si vs. P availability. Please do not cite this reference in such a way that was even not addressed by the authors of this paper. + as indicated above. 2,3) We agree with the reviewer, especially in the lack of physiological data used in the original study to assess a threshold silicic acid; please see reply to general comment. We have revised by using more specific language "e.g. based on correlating microplankton abundance and nutrient concentration, opposed to physiological analyses, a 2 $\mu$M threshold [Si(OH)4] has been used extensively to define where diatoms are outcompeted by flagellates (Egge and Aksnes, 1992)."

1) line 137: ". . . suggesting that N was likely more important than P for primary production." – As authors refer to absolute concentrations, the correct phrasing should be: ". . . suggesting that N was likely more important than P for potentially limiting primary production." 2,3) Changed as suggested.

1) line 138: "These phosphate data are not discussed." – Even though a range would be welcome. 2,3) Revised (e.g. "These phosphate data (0.1 – 0.6 $\mu$M in the upper 50 m) are not discussed.")

1) line 153: ". . . fixed with an aldehyde mixture of hexamethylenetetramine-buffered formaldehyde and glutaraldehyde at 0.1 and 1% final concentration, respectively, as suggested by Tsuji and Yanagita (1981) . . ." – although this should be OK this is not the usual fixative for diatoms (acidic Lugol preferred), partly due to its toxicity for the microscopical examiner. 2,3) Agreed. Given the interdisciplinary nature of the cruise and the lack of excess operation time, this fixative was used by the Norwegian Polar Institute group to do multiple analyses from one sample (which cell counts were only one); these other analyses are beyond the scope of this communication.

1) line 164: " . . . neutral density screened bags . . ." – please mention the photometric levels used. 2,3) Added ". . . neutral density screened bags simulating 50%, 20% and 1% of irradiance just below the surface."

1) line 207: "Export rates were calculated using the standing stock measurements, length of deployment, and trap opening area." – Please give the model/type of sediment trap. 2,3) Added "KC Denmark design (outer diameter 72 mm, length 450 mm)."

1) line 260: ". . . except for the Hinlopen ice algae, where the melt water . . ." – Is that naturally– melted ice or meltwater produced by ice melting in the lab? Please clarify. 2,3) Thank you for the question, this has been clarified, e.g. ". . . where water, which was melted at ambient air temperature on the vessel, had . . ."

1) Line 331: " Brown et al., 2003 " – Comment: For some strange reason L. Brown's

incubations lasted for only 6 hours, which renders her production results questionable. 2,3) We agree with the reviewer; however, given the limited data from the Northeast Atlantic, we prefer to be thorough and include the Brown et al. publication here.

1) line 339: "... Varela et al. (2013) recently reported that [Si(OH)4] in surface waters (>5 $\mu$M) are unlikely to be significantly limiting to diatoms in any sector of the Bering, Chukchi or Beaufort Sea regions." – Although for Subarctic waters Brown et al. (2003, mentioned just above) kinetic experiments show a strong limitation (non-saturating kinetics) up to 30 $\mu$M. 2,3) We agree with the reviewer; however, the goal of this sentence is to isolate the Canadian- & United-States Arctic waters with the European Arctic waters. We add the Brown et al. (2003) kinetic data in a section below (e.g. with Allen et al. 2005, Kristiansen et al. 2001 discussion near lines 442, 443).

1) line 387: " Suboptimal silicon availability affects the rate of diatom bSiO2 production and can limit their growth. A widely cited [Si(OH)4] threshold, below which diatoms will be outcompeted by other phytoplankton, is ~2.0 $\mu$M; this metric was derived from a comparison of diatom abundance (relative to total microplankton) versus [Si(OH)4] during mesocosm experiments in a Norwegian fjord system (Egge and Aksnes, 1992)." – should be removed: No need to discuss this threshold as it is mentioned that it is strongly criticized (and see my comment above). 2,3) Thank you for the perspective; this section will be removed (please see response to earlier comments on this topic).

1) line 410: " This indeed indicates that phytoplankton can deplete nitrogen to levels below detection while they appear unable to deplete Si(OH)4 pools below 0.5 $\mu$M, which would indicate 0.5 $\mu$M is the ultimate Si(OH)4 concentration required to support diatom growth." – I disagree with this interpretation. The 0.5 $\mu$M Si level just reflects the residual H4SiO4 stock after complete removal of nitrate. line 422: "... if diatoms are limited by an absolute [Si(OH)4] (e.g. 2 $\mu$M), ..." – This is speculative: By what evidence is this proposition supported? 2,3) Given the reviewer's comments for the paragraphs starting in line 405 (i.e. line 410 comment) and line 41 (line 422 comment), these paragraphs have been revised to demonstrate that assessment of nitrate vs.

[Figure]

silicic acid consumption relationships can be used to infer either N or Si limitation of diatom biomass and growth; this sets up the importance of infers growth dynamics using more direct physiological uptake data (e.g. Si uptake kinetics).

1) line 436: ". . . the relationship between Vb and [Si(OH)4] also supports that Si regulates diatom productivity to some degree." – The large dispersion of data points on Figure 5 results in a very weak relationship, so that there is certainly something else explaining the low realized Vb at the 4.5 $\mu$M H4SiO4 level. 2,3) The reviewer is correct, there is variability. But please note these data are from all diatom assemblages sampled among all stations and depths; thus, the linearity of the relationship would not be expected to be high, in fact, we were surprised to see nearly 40% of the variability explained given so many assemblages. However, we feel the Fig. 5 data, which integrates all samples, is the most conservative way to estimate the aggregate Ks value for euphotic diatoms among all stations at the time of this cruise.

1) line 443: " . . . Allen et al. (2005) observed a linear response in Vb between ambient and 5 $\mu$M [Si(OH)4], which suggests uptake did not show any degree of saturation at this concentration." – Also in Brown et al. (2003); as mentioned above. 2,3) The Brown et al. (2003) reference has been added.

1) line 517: " At van Mijenfjorden, the rate of export in the upper 40 m represented 39% of the 2 standing stock (23.3 mmol Si m-2) in the same vertical layer." – I don't understand as from Table 1 it seems that the standing stock is 10.8 and the export 9.03? 2,3) Thank you for the observation, this is clarified. We now reference Fig. 2C as the source of the van Mijenfjorden integral data. Table 1 integrates biogenic silica stock and production to 20 m (i.e. deepest depth among all profiles) whereas the shallows depth for sediment traps among all deployments was from 40 m. For van Mijenfjorden, we have biogenic silica measurements down to 50 m (Fig. 2C), which allows for the comparison as described in line 517.

1) line 524: " The rate of bSiO2 export was also at

least a factor of four higher than $\int$ in the upper 20m." – $-I\,was\,not\,able\,to\,find\,where\,did\,this\,come\,from.\,2,3)\,Thank\,you\,for\,the\,observation,\,this\,is\,a\,general\,trend\,among\,stations\,in\,Table$ $and\,production\,stations\,was\,also\,at\,least\,a\,factor\,of\,four\,higher\,than\,\int\,in\,the\,upper\,20m\,(Table\,1)."$

$Please\,also\,note\,the\,supplement\,to\,this\,comment:$
$https://www.biogeosciences-discuss.net/bg-2018-226/bg-2018-226-AC1-$
$supplement.pdf$

---

## Author Comment (AC2) · 20 Aug 2018

Below, we address comments from RC2. We cluster each comment and separate them as "1" Reviewer comments, "2,3" are responses and revisions. A pdf version (bg-2018-226-RC2-supplement_Krause_etal_response) of this response has been uploaded in the supplement.

1) Krause et al. investigated phytoplankton, especially diatoms, and nutrients at 9 stations in the Atlantic sector north of 76oN. They measured silicate, nitrate plus nitrite, chlorophyll a, biogenic silica, determined di- atom assemblage, estimate productivity and export (based on sediment traps). The silicic acid concentration in the upper 50 m was always below 5 $\mu$mol L-1 and at most stations below the nitrate plus nitrite concentration. At several stations [Si(OH)4] was below 1 or even 0.5 $\mu$mol L-1 in the upper 20 m. In order to investigate Si uptake limitation, the authors performed on board growth experiments over a range of [Si(OH)4] at 4 stations. Michaelis-Menten functions for sili-cic acid uptake (Eq.1) were fit to the data yielding estimates for maximum uptake rates (Vmax) and half- saturation constants (KS). Let me suggest listing these estimates in a table. The estimates for KS are much higher than some estimates (for different diatom species) reported in the literature (for example, Paasche, 1973a,b), however, lower than the higher values given by Kristiansen et al. (2000). What might explain this large range and these differences? Could it be influenced by factors (other nutrients, graz-ing) differing between the various investigations/experiments? 2,3) We speculate that diversity and diatom origin (e.g. more Atlantic influenced waters, perhaps residual ice diatoms) may be some of the underlying factors. However, these are (unfortunately) beyond the scope of our data. What is important, at least for a modeling perspective, is that these kinetic parameters are published and available to ground truth regional simulations.

1) The manuscript contains valuable new data, is well written, and will be of interest to many readers of Biogeosciences. I recommend publication after minor revisions. 2,3) We thank the reviewer for this assessment and reply to the revisions below.

Further remarks/suggestions: 1) L123-129 Description of trap deployment was not very detailed ('at 3-7 depths between 20 and 150-200 m, based on bathymetry'). It would be good to add a list with depths and bottom depths (or a reference where to find this information). 2,3) This has been added to the prose. This ranged from shallow ($\sim$60 m, van Mijenfjorden) to deep (260-290m in Erik Erikssenstretet and Atlantic stations, respectively).

1) L132-134 Freezing sample for nutrient analysis: Procedures (thawing, measure-ments how long after thawing) and quality control (freeze certified reference material in parallel with samples) have been discussed in the literature (for example, Macdonald et al., 1986, Clementson & Wayte, 1992, Dore et al., 1996). Could you please give

more details on the procedures and quality control? 2,3) Co-Author Kristiansen's laboratory has extensive experience in these analyses. Pertinent details have been added, including reference seawater from Ocean Scientific International Ltd. (UK, Line 135), and detection limits (Line 136) are available in the initial submission. Standard practices (slow thawing of silicic acid samples to allow depolymerization, three parallels measured, etc.), have been added along with prose regarding the analytical reproducibility (median coefficient of variation was 5% for NO3+NO2 and PO4, 2% for silicic acid and 9% for NO2 → higher coefficient of variation was observed when the absolute concentrations were low, e.g. <0.1 uM, hence using the median value). Because of the cruise duration and transfers, and the well-known issues of getting reliable measurements from frozen samples, no ammonium was measured.

1) L136-138 "Phosphate was analyzed, but N:P ratios for nutrients were, on average, 8 among all stations, suggesting that N was likely more important than P for primary production." N:P is below Redfield thus N might be limiting primary production before P. However, 'N was likely more important than P for primary production' sounds strange. Please rewrite. 2,3) This has been modified. "Phosphate was analyzed, but N:P ratios for nutrients were, on average, 8 among all stations; this suggests N was more likely than P to limit primary production."

1) L145,148,225 60o C → 60oC, -20 oC → -20oC, -2-1 oC → -2 to +1oC (no gaps; please check whole manuscript) 2,3) Gaps have been removed through the whole manuscript.

1) L175 please rewrite "dividing by the depth integration" → dividing by depth-integrated values 2,3) This has been modified as suggested.

1) L199 ml → mL 2,3) This has been modified.

1) L205-206 using C:Si (instead of Si:C) would avoid the exponent -1 in Eq.(2) and give values more in Redfield-style, i.e. molar Si:C = 0.13 → 7.7 C:Si (only slightly higher than the Redfield C:N). What's the uncertainty of the Si:C estimate? 2,3) While

C:Si and Si:C can both be used, we chose the Si:C based on convention established by other publications when making these types of estimates (e.g. Nelson et al. 1995 GBC, Nelson and Brzezinski 1997 L&O, Leynaert et al. 2001 DSR I, Brzezinski et al. 2011 DSR I, Krause et al. 2011 DSR II, Krause et al. 2015 JGR Oceans). Regarding uncertainty, please see response to reviewer 1. This uncertainty in Si:C, even if a factor of two, does not change the two main interpretations (1, diatoms "bloom and bust," 2, even when diatom biomass is relatively low their contribution to primary production is quantitatively important).

1) L295-296 "The rate of diatom biogenic silica production was reduced by ambient [Si(OH)4] in 95% of the samples examined." sounds strange. I guess you mean 'was kinetically limited by ambient [Si(OH)4]' based on comparison with estimated KS values or based on enhancement factors. 2,3) This has been modified.

1) L317,548 Spearman's Rho Test: add number of data n = ... 2,3) This has been added (n = 15).

1) L380-384 What about grazing? 2,3) Grazing would affect the standing stock of diatom biomass (and thus the absolute rate of production, Rho), but not the specific rates (e.g. VAVE) which are more likely driven by growth/bottom up factors. However, in this region, grazing is likely the primary mechanism which transforms living diatom silica into detrital silica. Because the latter is a minor and speculative point given the data, we feel adding a complicated explanation about grazing here would stymie the narrative flow without adding enough clarity.

References [1] Clementson, Lesley A. and Wayte, Sally E. The effect of frozen storage of open-ocean seawater samples on the concentration of dissolved phosphate and nitrate. Water Research, 26(9):1171–1176, 1992.

[2] Dore, J.E., T. Houlihan, D.V. Hebel, G. Tien, L. Tupas, and D.M. Karl. Freezing as a method of sample preservation for the analysis of dis- solved inorganic nutrients in seawater. Marine Chemistry, 53(3-4):173– 185, 1996.

[3] Macdonald, RW and McLaughlin, FA and Wong, CS. The storage of reactive silicate samples by freezing. Limnology and Oceanography, 31(5):1139–1142, 1986.

[4] Paasche, E. Silicon and the ecology of marine plankton diatoms. I. Tha- lassiosira pseudonana (Cyclotella nana) grown in a chemostat with silicate as limiting nutrient. Marine biology, 19(2):117–126, 1973a.

[5] Paasche, E. Silicon and the ecology of marine plankton diatoms. II. Silicate-uptake kinetics in five diatom species. Marine Biology, 19(3):262–269, 1973b.

Please also note the supplement to this comment:
https://www.biogeosciences-discuss.net/bg-2018-226/bg-2018-226-AC2-supplement.pdf

——————————————————

---

## Author Response (AR1)

Below, we address comments from Reviewer 1 (RC1), and Reviewer 2 (RC2). We cluster each comment and separate them as "1" Reviewer comments, "2,3" are responses and revisions.

REVIEWER 1

1) The manuscript of Krause *et al.* presents interesting results concerning biogenic silica production and export levels as well as estimates of kinetic constants from an opportunistic sampling near the Svalbard Archipelago in the Arctic Ocean by late spring. The data presented are the first direct (silicon–32 method) measurements of biogenic silica production in the Arctic Ocean, which in themselves deserve publication. From these data, the authors then attempt to establish the potential control of diatom production by the availability of silicic acid as well as the contribution of diatoms to total primary production. However, I think the authors are pushing their limited data set much too far and that the manuscript should be shortened by getting more concise.

2,3) We thank the reviewer for the constructive criticism, these have strengthened the manuscript. Below, we address the concerns. The revisions are aimed to both address these issues and make the manuscript more concise.

1) Firstly, I find it difficult to understand why the authors focus on the so-called Egge & Aksnes 2 µM H4SiO4 threshold value, as it is clear that the data from this publication have been wrongly interpreted in several past publications (which is recognized by the authors besides).

2,3) We agree with the reviewer and the focus on this apparent threshold value is due to its wide use in the field. As of August 2018, the Egge & Aksnes (1992) manuscript has been cited over 700 times, and many explicitly regard this 2 µM threshold for diatom microplankton dominance among many systems. This type of prose has been reduced (now just in the discussion (line 480); however, given this is the threshold embraced by the community it must be acknowledged.

1) I suggest just using their kinetic values to discuss the potential limitation of diatom uptake by H4SiO4 availability, and then shortly discuss hypotheses for growth limitation, which is another point not directly assessed in this study. ON the other hand, authors might consider that the actual limitation starts under 2 times $K_S$.

2,3) The original version discusses kinetic limitation between lines 434 to 455, followed by a section discussing the possibility that silicic acid may be low enough to limit diatom growth (line 459 – 462). If we interpret the reviewer correctly, then "under 2 times Ks" (i.e. half of the half-saturation constant) is the silicic acid concentration when uptake (i.e. Vb) is 25% of maximum uptake (i.e. Vmax) which is the same metric we use to diagnose potential growth limitation (line 461). We have made this clearer (e.g. denote the equivalence of "under 2 times Ks" and Vb/Vmax <0.25) in the revision (line 473-474).

1) The comparison between nitrate and silicic acid concentrations is not very clear. First, they mention a 2.5 slope (Figure 5) between nitrate and silicic acid, which means that nitrate is taken up 2.5 times faster than silicic acid, and then authors take a 1:1 ratio to discuss the potential for silicic acid limitation. I can imagine that they are trying to decipher the relative contribution of siliceous vs. non-siliceous (e.g. *Phaeocystis*) components of the phytoplankton community but this should be clearly indicated. On top of that, the use of a 1:1 Si:N ratio is questionable (large species–specific variations, see below for Si:C).

2,3) The approach silicic acid versus nitrate drawdown has been used elsewhere (Monterey

Bay, Brzezinski et al. 1997; Barents Sea, Rey et al. 1987; Southern Ocean diatom cultures, Takeda 1998). We agree with the reviewer that this nutrient data includes drawdown of nitrate from other organisms, likely *Phaeocystis*, hence the slope exceeds 1 (i.e. nitrate drawn down faster than silicic acid). Most polar culture work focuses on Fe-limitation (e.g. Takeda 1988, Nature) and Southern Ocean clones; however, under Fe-replete conditions Takeda (1998) reported Si:N between 0.73 (*Chaetoceros dichaeta*) and 1.2 (*Nitzschia* sp.), the average among these clones is 0.97, not unlike the 1:1 ratio from Brzezinski 1985. In the revision, we make this point with more clarity and provide the Takeda (1998) study as a literature basis for the 1:1 ratio (beyond the canonical Brzezinski stoichiometry) and also include new culture data in review (Lomas et al., which Krause is co-author on this manuscript) which examined 11 polar diatom clones (lines 416-424).

1) The contribution of diatoms to primary production is also another weak point of the ms. The calculation is based on a transformation of rSi to rC by using the average Brzezinski' ratio of 0.13. This is a very simplistic way of addressing this important question as this ratio is known to be subject to large species–specific variations (e.g. Brzezinski gives a Si:C biomass range of 0.15 ±0.04 for large diatoms, which could result in a ~2 times range for rC estimates). This is somehow risky business and should, at least, be acknowledged and discussed.
2,3) As with the previous point, we agree that there is variability within the stoichiometry. While admittedly simplistic, we also have coarse resolution (n = 6 coupled measurements), which (in our opinion) do not merit as sophisticated approach given it would not be robust; this data coupling was serendipitous (hence the low n value) but does provide a first-order estimate. We have used the Lomas et al. (in review) diatom Si:C ratios (see methods, lines 218-223), which changed the magnitude of the estimates but not the general interpretations: 1) diatoms quantitative contribution to primary production is "boom and bust" and 2) even when diatom biomass is relatively low, their contribution to production can still be quantitatively important (e.g. 25%). Additionally, the new analysis is potentially conservative (line 510) given any movement of the Si:C ratio used toward the canonical Brzezinski ration increases the proportion of production attributed to diatoms.

1) Finally, the authors present data for direct diatom cell export but the underlying issue is not clearly stated: Do they want to compare direct diatom sedimentation by mass sinking to other export vectors such as repackaging? If yes this should be clearly stressed.
2,3) The reviewer's interpretation of our intent is correct, we suggest that the reason why cellular export is similar to previous studies, despite biogenic silica export being higher than other studies, is consistent with repackaging which we termed "food web effect." We have revised this for clarification (paragraph starting on line 577).

1) line 28 : "diatom cellular export" – the wording is misleading (could be export from a diatom cell). I'd rather use "export of diatom cells".
2,3) Changed as suggested, here and in other places.

1) line 65 : " A more recent analysis demonstrated a decline in pre-bloom [Si(OH)4] concentrations by 1–2 µM across the north Atlantic subpolar and polar regions over the last 25 years (Hátún *et al.*, 2017); this is consistent with the general Arctic region being a net exporter of silicic acid (Torres-Valdés *et al.*, 2013)." – I don't see the consistence between the decrease of H4SiO4 concentrations and the net exportation of this nutrient; please rephrase.
2,3) This is rephrased, e.g. removing "this is consistent with the general Arctic region being a net exporter of silicic acid (Torres-Valdés *et al.*, 2013)" (now line 67).

1) line 68: " This is in stark contrast to the 10–60 µM [Si(OH)4] observed in the surface waters of the Southern Ocean and the marginal ice zone around Antarctica (Nelson and Gordon, 1982; Brzezinski et al., 2001), where [Si(OH)4] is unlikely to limit the rate of diatom production or biomass yield." – I disagree; There are ample references to state that actually the reverse is true, due to sometimes unusual high $K_S$ (e.g. Nelson & Tréguer MEPS 1992, Nelson et al. DSR II 2001, Mosseri et al. DSR II 2008).

2,3) The reviewer is correct as kinetic limitation (i.e. ambient silicic acid limits the rate of diatom silica production) is clearly observed in the Southern Ocean. Our intent (which was not clearly conveyed) was to contrast the effects of the high silicic acid in the Southern Ocean vs. the Arctic. This is rephrased "… where [Si(OH)4] is unlikely to limit diatom growth unless iron is replete, and stimulates exceptional blooms which consume Si, or assemblages are highly inefficient for Si uptake." (line 69-71).

1) line 76: " … and a 2 µM threshold [Si(OH)4] defines where diatoms are outcompeted by flagellates (Egge and Aksnes, 1992)." – I strongly disagree with that sentence. The work of Egge and Aksnes did not evidence any real threshold (no kinetic values measured) and just merely indicated areas of realized niches for diatom vs. flagellates with regards to Si vs. P availability. Please do not cite this reference in such a way that was even not addressed by the authors of this paper. + as indicated above.

2,3) We agree with the reviewer, especially in the lack of physiological data used in the original study to assess a threshold silicic acid; please see reply to general comment. We have revised by using more specific language "Egge and Aksnes (1992) data set shows diatoms may be outcompeted by flagellates when [Si(OH)4] <2 µM, a value which is more reflective of an ecological niche opposed to a physiological threshold as has been purported in numerous citations of these data." (line 480-483).

1) line 137: "… suggesting that N was likely more important than P for primary production." – As authors refer to absolute concentrations, the correct phrasing should be: "… suggesting that N was likely more important than P for potentially limiting primary production."

2,3) Changed as suggested.

1) line 138: "These phosphate data are not discussed." – Even though a range would be welcome.

2,3) Revised line 145 (e.g. "These phosphate data (0.1 – 0.6 µM in the upper 50 m) are not discussed.")

1) line 153: "… fixed with an aldehyde mixture of hexamethylenetetramine-buffered formaldehyde and glutaraldehyde at 0.1 and 1% final concentration, respectively, as suggested by Tsuji and Yanagita (1981) …" – although this should be OK this is not the usual fixative for diatoms (acidic Lugol preferred), partly due to its toxicity for the microscopical examiner.

2,3) Agreed. Given the interdisciplinary nature of the cruise and the lack of excess operation time, this fixative was used by the Norwegian Polar Institute group to examine a wide range of protist groups from one sample; these other protist groups are beyond the scope of this communication.

1) line 164: " … neutral density screened bags …" – please mention the photometric levels used.

2,3) Added "… neutral density screened bags simulating 50%, 20% and 1% of irradiance just below the surface." (line 171)

1) line 207: "Export rates were calculated using the standing stock measurements, length of deployment, and trap opening area." – Please give the model/type of sediment trap.
2,3) Added "KC Denmark design (outer diameter 72 mm, length 450 mm)." (line 127)

1) line 260: "… except for the Hinlopen ice algae, where the melt water …" – Is that naturally– melted ice or meltwater produced by ice melting in the lab? Please clarify.
2,3) Thank you for the question, this has been clarified, e.g. "… water, which was melted at ambient air temperature on the vessel …" (line 277).

1) Line 331: " Brown *et al.*, 2003 " – Comment: For some strange reason L. Brown's incubations lasted for only 6 hours, which renders her production results questionable.
2,3) We agree with the reviewer; however, given the limited data from the Northeast Atlantic, we prefer to be thorough and include the Brown et al. publication here.

1) line 339: "… Varela *et al.* (2013) recently reported that [Si(OH)4] in surface waters (>5 µM) are unlikely to be significantly limiting to diatoms in any sector of the Bering, Chukchi or Beaufort Sea regions." – Although for Subarctic waters Brown *et al.* (2003, mentioned just above) kinetic experiments show a strong limitation (non-saturating kinetics) up to 30 µM.
2,3) We agree with the reviewer; however, the goal of this sentence is to isolate the Canadian- & United-States Arctic waters with the European Arctic waters. We add the Brown et al. (2003) kinetic data in a section below (e.g. with Allen et al. 2005, Kristiansen et al. 2001 discussion in the paragraph starting at line 445).

1) line 387: " Suboptimal silicon availability affects the rate of diatom bSiO2 production and can limit their growth. A widely cited [Si(OH)4] threshold, below which diatoms will be outcompeted by other phytoplankton, is ~2.0 µM; this metric was derived from a comparison of diatom abundance (relative to total microplankton) versus [Si(OH)4] during mesocosm experiments in a Norwegian fjord system (Egge and Aksnes, 1992)."
– should be removed: No need to discuss this threshold as it is mentioned that it is strongly criticized (and see my comment above).
2,3) Thank you for the perspective; this section will be removed (please see response to earlier comments on this topic).

1) line 410: " This indeed indicates that phytoplankton can deplete nitrogen to levels below detection while they appear unable to deplete Si(OH)4 pools below 0.5 µM, which would indicate 0.5 µM is the ultimate Si(OH)4 concentration required to support diatom growth."
– I disagree with this interpretation. The 0.5 µM Si level just reflects the residual H4SiO4 stock after complete removal of nitrate.
line 422: "… if diatoms are limited by an absolute [Si(OH)4] (e.g. 2 µM), …" – This is speculative: By what evidence is this proposition supported?
2,3) Given the reviewer's comments for the paragraphs starting in line 405 (i.e. line 410 comment) and line 41 (line 422 comment), these paragraphs have been revised (now line 441 – line 444).

1) line 436: "… the relationship between Vb and [Si(OH)4] also supports that Si regulates diatom productivity to some degree." – The large dispersion of data points on Figure 5

results in a very weak relationship, so that there is certainly something else explaining the low realized $V_b$ at the 4.5 µM H4SiO4 level.

2,3) The reviewer is correct, there is variability. But please note these data are from all diatom assemblages sampled among all stations and depths; thus, the linearity of the relationship would not be expected to be high, in fact, we were surprised to see 71% of the variability explained (when Hornsunddjupet excluded) given so many assemblages. However, we feel the Fig. 5B data, which integrates all samples, is the most conservative way to estimate the aggregate Ks value for euphotic diatoms among all stations at the time of this cruise. We also note that the incorrect slope and R2 were plotted in the original version of Fig 5B (now corrected).

1) line 443: " … Allen *et al.* (2005) observed a linear response in Vb between ambient and 5 µM [Si(OH)4], which suggests uptake did not show any degree of saturation at this concentration." – Also in Brown *et al.* (2003); as mentioned above.

2,3) The Brown et al. (2003) reference has been added here, as indeed, it is the original source of the Allen et al. 2005 Si-uptake data (e.g. line 454).

1) line 517: " At van Mijenfjorden, the rate of export in the upper 40 m represented 39% of the ∫bSiO2 standing stock (23.3 mmol Si m-2) in the same vertical layer." – I don't understand as from Table 1 it seems that the standing stock is 10.8 and the export 9.03?

2,3) Thank you for the observation, this is clarified. We now reference Fig. 2C as the source of the van Mijenfjorden integral data (line 542). Table 1 integrates biogenic silica stock and production to 20 m (i.e. deepest depth among all profiles) whereas the shallowest depth for sediment traps among all deployments was from 40 m. For van Mijenfjorden, we have biogenic silica measurements down to 50 m (Fig. 2C), which allows for the comparison.

1) line 524: " The rate of bSiO2 export was also at least a factor of four higher than ∫ρ in the upper 20 m." – I was not able to find where did this come from.

2,3) Thank you for the observation, this is a general trend among stations in Table 1 (i.e. biogenic silica export was much higher than integrated biogenic silica production). This has been revised "The rate of bSiO2 export among all export- and production stations was also at least a factor of four higher than ∫ρ in the upper 20 m (Table 1)." (line 549, 550).

Below, we address comments from RC2. We cluster each comment and separate them as "1" Reviewer comments, "2,3" are responses and revisions. A pdf version (bg-2018-226-RC2-supplement_Krause_etal_response) of this response has been uploaded in the supplement.
* * *
REVIEWER 2

1) Krause et al. investigated phytoplankton, especially diatoms, and nutrients at 9 stations in the Atlantic sector north of $76^o$N. They measured silicate, nitrate plus nitrite, chlorophyll a, biogenic silica, determined di- atom assemblage, estimate productivity and export (based on sediment traps). The silicic acid concentration in the upper 50 m was always below 5 $\mu$mol L$^{-1}$ and at most stations below the nitrate plus nitrite concentration. At several stations [Si(OH)$_4$] was below 1 or even 0.5 $\mu$mol L$^{-1}$ in the upper 20 m. In order to investigate Si uptake limitation, the authors performed on board growth experiments over a range of [Si(OH)$_4$] at 4 stations. Michaelis-Menten functions for silicic acid uptake (Eq.1) were fit to the data yielding estimates for maximum uptake rates ($V_{max}$) and half- saturation constants ($K_S$). Let me suggest listing these estimates in a table. The estimates for $K_S$ are much higher than some estimates (for different diatom species) reported in the literature (for example, Paasche, 1973a,b), however, lower than the higher values given by Kristiansen et al. (2000). What might explain this large range and these differences? Could it be influenced by factors (other nutrients, grazing) differing between the various investigations/experiments?

2,3) We speculate that diversity and diatom origin (e.g. more Atlantic influenced waters, perhaps residual ice diatoms) may be some of the underlying factors. However, these are (unfortunately) beyond the scope of our data. What is important, at least from a modeling perspective, is that these kinetic parameters are published and available to ground truth regional simulations.

1) The manuscript contains valuable new data, is well written, and will be of interest to many readers of Biogeosciences. I recommend publication after minor revisions.

2,3) We thank the reviewer for this assessment and reply to the revisions below.

**Further  remarks/suggestions**:

1) L123-129 Description of trap deployment was not very detailed ('at 3-7 depths between 20 and 150-200 m, based on bathymetry').  It would be good to add a list with depths and bottom depths (or a reference where to find this information).

2,3) This has been added to the prose (paragraph starting at line 122).

1) L132-134 Freezing sample for nutrient analysis: Procedures (thawing, measurements how long after thawing) and quality control (freeze certified reference material in parallel with samples) have been discussed in the literature (for example, Macdonald et al., 1986, Clementson & Wayte, 1992, Dore et al., 1996). Could you please give more details on the procedures and quality control?

2,3) Co-Author Kristiansen's laboratory has extensive experience in these analyses. Pertinent details have been added (lines 132-145) beyond the reference seawater from Ocean Scientific International Ltd. (UK) and detection limits are available in the initial submission. Standard practices (slow thawing of silicic acid samples to allow depolymerization, three parallels measured, etc.), have been added (and suggested references) along with prose regarding the analytical reproducibility (median coefficient of variation was 5% for NO3+NO2 and PO4, 2% for silicic acid and 9% for NO2 → higher coefficient of variation was observed when the absolute concentrations were low, e.g. <0.1 uM, hence using the median value). Because of the cruise duration and transfers, and the well-known issues of getting reliable measurements from frozen samples, no ammonium was measured (also clarified in revision).

1) L136-138 "Phosphate was analyzed, but N:P ratios for nutrients were, on average, 8 among all stations, suggesting that N was likely more important than P for primary production." N:P is below Redfield thus N might be limiting primary production before P. However, 'N was likely more important than P for primary production' sounds strange. Please rewrite.

2,3) This has been modified.  "Phosphate was analyzed, but N:P ratios for nutrients were, on average, 8 among all stations; suggesting that N was likely more important than P for potentially limiting primary production. These phosphate data (0.1–0.6 µM in the upper 50 m) are not discussed." (line 143-145)

1) L145,148,225 $60^{o}$C $\rightarrow$ $60^{o}$C, $-20^{o}$C $\rightarrow$ $-20^{o}$C, $-2$-$1^{o}$C $\rightarrow$ $-2$ to $+1^{o}$C (no gaps; please check whole manuscript)
2,3) Gaps have been removed through the whole manuscript.

1) L175 please rewrite "dividing by the depth integration" $\rightarrow$ dividing by depth-integrated values
2,3) This has been modified as suggested.

1) L199 ml $\rightarrow$ mL
2,3) This has been modified.

1) L205-206 using C:Si (instead of Si:C) would avoid the exponent -1 in Eq.(2) and give values more in Redfield-style, i.e. molar Si:C = 0.13 $\rightarrow$ 7.7 C:Si (only slightly higher than the Redfield C:N). What's the uncertainty of the Si:C estimate?
2,3) While C:Si and Si:C can both be used, we chose the Si:C based on convention established by other publications when making these types of estimates (e.g. Nelson et al. 1995, Nelson and Brzezinski 1997, Leynaert et al. 2001, Brzezinski et al. 2011, Krause et al. 2011, Krause et al. 2015). Regarding uncertainty, please see response to RC1.

1) L295-296 "The rate of diatom biogenic silica production was reduced by ambient [$Si(OH)_4$] in 95% of the samples examined." sounds strange. I guess you mean 'was kinetically limited by ambient [$Si(OH)_4$]' based on comparison with estimated $K_S$ values or based on enhancement factors.
2,3) This has been modified.

1) L317,548 Spearman's Rho Test: add number of data n = ...
2,3) This has been added (n = 15).

1) L380-384 What about grazing?
2,3) Grazing would affect the standing stock of diatom biomass (and thus the absolute rate of production, Rho), but not the specific rates (e.g. $V_{AVE}$) which are more likely driven by growth/bottom up factors. However, in this region, grazing is likely the primary mechanism which transforms living diatom silica into detrital silica. Because the latter is a minor and speculative point given the data, we feel adding a complicated explanation about grazing here would stymie the narrative flow without adding enough clarity.

1) Clementson, Lesley A. and Wayte, Sally E. The effect of frozen stor- age of open-ocean seawater samples on the concentration of dissolved phosphate and nitrate. *Water Research*, 26(9):1171–1176, 1992.
Macdonald, RW and McLaughlin, FA and Wong, CS. The storage of reactive silicate samples by freezing. Limnology and Oceanography, 31(5):1139–1142, 1986.
2,3) These two references were added (line 137-138); we thank the reviewer for the suggestion.
* * *
END OF RESPONSE

[revised manuscript text omitted]